# Never Forget the Basics: In-distribution Knowledge Retention for Continual Test-time Adaptation in Human Motion Prediction

## Abstract

This paper presents a novel approach to addressing the underexplored challenge of human pose prediction in dynamic target domains that simultaneously contain in-distribution (ID) and out-of-distribution (OOD) data. Existing test-time adaptation (TTA) techniques predominantly focus on OOD data, neglecting the fact that ID data, which closely resembles the training distribution, is often encountered during real-world deployment, leading to significant degradation in ID performance. To address this, we introduce In-Distribution Knowledge Retention (IDKR), a continual TTA framework designed to preserve critical knowledge about ID data while adapting to unseen OOD sequences. Our method introduces an ID-informative subgraph learning strategy that leverages the structural characteristics of human skeletal data to compute a structural graph Fisher Information Matrix (SG-FIM). Unlike prior work, IDKR simultaneously considers both node and edge features in the skeletal graph, with edge features, representing the invariant bone lengths between parent-child joint pairs, being essential for maintaining structural consistency across poses. These edge features are key to extracting reliable SG-FIM parameters, enabling the model to retain parameters critical for ID performance while selectively updating those needed for OOD adaptation. Extensive experiments on multiple benchmark datasets demonstrate that IDKR consistently outperforms state-of-the-art methods, particularly in scenarios involving mixed ID and OOD data, setting a new standard for robust human pose prediction in dynamic environments.

## 1 Introduction

3D human pose prediction (HPP) is a fundamental task in computer vision and machine intelligence, with applications in human-robot interaction, and robotics (Lou et al., 2024; Yan et al., 2024). The goal of HPP is to predict future human poses based on a sequence of observed 3D poses.

State-of-the-art HPP methods predominantly follow a data-driven deep learning approach, where models are trained on large-scale datasets and directly applied to unseen target data (Guo et al., 2023; Chen et al., 2023; Wang et al., 2023). However, a key challenge for these models is domain shift, where a model trained on a source domain performs poorly when deployed in a new, unseen target domain. Domain shifts occur due to differences in body shapes, proportions, and motion patterns between the training and test data (Liang et al., 2024; Zhang et al., 2023). To address this issue, recent studies (Cui et al., 2023b;a) have proposed TTA techniques, which dynamically adjust the pre-trained model during inference to better accommodate target domain shifts.

Despite these advancements, current TTA methods exhibit two notable limitations when applied to real-world HPP scenarios:

- **Unrealistic Stationary Target Domain Assumption:** Existing TTA-based HPP methods (Cui et al., 2023a;b) assume that the distribution shift between the source and target domains remains static, which is unrealistic in real-world dynamic environments.

- **In-Distribution and Out-of-Distribution Data in the Target Domain:** Current TTA methods (Wang et al., 2020; 2022; Brahma & Rai, 2023) do not adequately address the coexistence of in-distribution (ID) data, which resembles the training data, and out-of-

distribution (OOD) data, which differs from it. This results in suboptimal performance when both ID and OOD data are present in the target domain.

It is noteworthy that in real-world HPP deployments, the operational environment is dynamic, with individual behavioral patterns evolving over time, thus leading to a non-stationary target domain (Wang et al., 2022; Sanyal et al., 2023). This ongoing change of the target domain results in a perpetual distribution shift, which is not adequately addressed by existing TTA-based HPP methods. More importantly, in practical settings, alongside the OOD data, the deployment phase will inevitably encounter ID sequences that closely align with the distribution of the source domain (Sun et al., 2022; Li et al., 2022b; Mao et al., 2024), as evidenced in Appendix-B. However, existing methods often assume that the target domain is exclusively OOD, thus overlooking the presence of ID sequences, which limits their generalization performance.

To address these challenges, we propose a novel continual TTA framework, IDKR, which is specifically designed to handle evolving target domains that contain both ID and OOD data. Our framework not only enhances performance on OOD data but also retains key parameters critical for ID recognition, ensuring robust performance across both types of motion sequences. Our approach builds on the insight that human motion can be represented as a graph, where nodes correspond to human joints and edges represent skeletal connections. Recent advances in Graph Invariant Learning (GIL) have demonstrated the effectiveness of extracting informative subgraph structures for downstream tasks by filtering out irrelevant elements (Chen et al., 2024b; Li et al., 2022c). For example, in distinguishing between walking and running, the movements of the legs and arms provide more significant discriminatory information than the torso. Building on this, we integrate the Graph Information Bottleneck (GIB) framework (Wu et al., 2020) into GIL theory, proposing a novel method for in-distribution subgraph learning that compresses the original skeleton graph into an informative subgraph relevant to the ID labels (You et al., 2020). Once the ID-informative subgraph is identified, we incorporate this structure into the model's continual adaptation process. The in-distribution subgraph is used to guide the model's parameter updates during test-time adaptation. By aligning the model's focus on this subgraph, we ensure that the knowledge specific to ID data is preserved throughout the adaptation process, even as the model learns to handle OOD data.

Considering the skeletal topology of motion sequence, instead of the standard Fisher Information Matrix (FIM), we introduce an innovative ID-parameter estimation strategy–Structural Graph Fisher Information Matrix (SG-FIM), which quantifies the importance of model parameters based on both node and edge features within the ID-informative subgraph. The edge features, which represent the invariant bone lengths between parent and child joints, are crucial for maintaining structural consistency across different poses. This invariance is essential for accurately extracting SG-FIM, allowing the model to retain parameters critical for ID data while updating others for OOD adaptation. The SG-FIM serves as an ID knowledge retention regularization, integrated into the TTA optimization process via self-supervised loss (Tian & Lyu, 2024; Brahma & Rai, 2023), ensuring that ID-specific knowledge is preserved while the model adapts to the evolving target domain. This allows our approach to dynamically adapt to both ID and OOD data, achieving optimal predictive performance in a variety of deployment scenarios.

Our contributions can be summarized as follows: 1) We introduce the first framework that explicitly addresses the challenge of mixed ID and OOD target domains in HPP, overcoming a critical limitation of current TTA approaches. 2) We propose a novel IDKR that leverages ID-informative subgraph learning and SG-FIM to identify and retain ID-specific parameters while adapting to OOD data. 3) Extensive evaluations show that IDKR significantly outperforms state-of-the-art HPP models, particularly in non-stationary target domains containing both ID and OOD data.

## 2 RELATED WORKS

### 2.1 HUMAN POSE PREDICTION (HPP)

Early approaches to HPP primarily rely on RNN-based models, treating the task as a sequence-to-sequence generation problem by mapping historical poses to future predictions (Ruiz et al., 2018; Gui et al., 2018). While RNNs capture temporal correlations, they exhibit limitations such as static pose prediction and discontinuities between frames. To address these shortcomings, graph convolutional networks (GCNs) gain prominence due to their ability to model the semantic relationships within 3D skeleton structures (Dang et al., 2021; Li et al., 2022d; Chen et al., 2024a). Recent works

also emphasize the effectiveness of transformer-based architectures (Martínez-González et al., 2021; Aksan et al., 2021; Dai et al., 2023) and multilayer perceptrons (MLP) (Guo et al., 2023; Bouazizi et al., 2023) in capturing long-term dependencies, offering greater flexibility in motion prediction.

Despite these advancements, most methods operate under the assumption that training and test data come from the same distribution, which is unrealistic in real-world applications where target domains often exhibit domain shifts (Brahma & Rai, 2023; Yuan et al., 2023; Zhao et al., 2023). To address this discrepancy, recent studies (Cui et al., 2023b;a) introduce TTA techniques, which are constrained by their assumption of a stationary target domain. As a result, they struggle to handle continuously evolving environments or mixed ID and OOD samples within the target domain. Our approach explicitly tackles these limitations by providing a framework capable of adapting to both ID and OOD data in dynamic settings, improving the model's generalization across domains.

## 2.2 TEST-TIME ADAPTATION (TTA) AND CONTINUAL TTA (CTTA)

TTA represents a widely adopted source-free domain adaptation technique (Wang et al., 2020; Liu et al., 2021a; Gong et al., 2022a; Su et al., 2023; Zhao et al., 2023; Sreenivas et al., 2024), enabling pre-trained models to fine-tune themselves on individual test samples, thereby customizing their predictions post-deployment. In the context of HPP, TTA-based approaches (Cui et al., 2023a;b) have achieved promising results. However, standard TTA often assumes that the target distribution is static, which is unrealistic in dynamic real-world HPP settings.

Continual TTA (CTTA) extends TTA by allowing the model to adapt incrementally to dynamic target domains (Gong et al., 2022b; Sójka et al., 2023; Sanyal et al., 2023; Tian & Lyu, 2024). Methods like CoTTA (Wang et al., 2022) use a teacher-student framework with random restoration to mitigate catastrophic forgetting, while PETAL (Brahma & Rai, 2023) employs probabilistic modeling to stabilize updates during inference. However, most existing CTTA methods overlook the coexistence of ID and OOD data in the target domain, leading to performance degradation on ID sequences, particularly during long-term adaptation. Our method addresses this gap by explicitly identifying and preserving ID-specific parameters during adaptation, ensuring sustained performance on ID data throughout continual adaptation processes.

## 2.3 OUT-OF-DISTRIBUTION (OOD) DETECTION

OOD detection is crucial for distinguishing ID from OOD data in the target domain. These methods can be broadly classified into supervised and unsupervised categories, with supervised approaches being particularly relevant in our case, as all source data are labeled as ID during training (Sun et al., 2022; Li et al., 2022b; Mao et al., 2024). However, identifying ID samples in graph-structured data, such as human skeletons, presents unique challenges for conventional OOD detection techniques.

To address it, we propose a GIL-based OOD detection method that leverages the inherent graph structure of human skeletons. GIL allows us to compress subgraphs based on ID-specific information (Li et al., 2022c; Chen et al., 2024b). Unlike traditional OOD detection methods, which typically require explicit OOD labels, our approach does not depend on these labels. Instead, we focus on extracting ID-informative subgraphs that guide the subsequent identification and retention of ID-specific parameters, ensuring more robust performance across both ID and OOD data.

## 2.4 GRAPH INVARIANT LEARNING (GIL)

GIL aims to capture invariant relationships between graph features and labels while filtering out spurious correlations. Recent research in GIL, particularly in causal learning (Wu et al., 2022; Liu et al., 2022) and graph manipulation (Li et al., 2022c; 2023), has demonstrated its effectiveness in OOD generalization and detection. In particular, the Graph Information Bottleneck (GIB) framework has been shown to learn robust graph representations by optimizing mutual information between subgraphs and their corresponding labels (Wu et al., 2020; Wang et al., 2024).

Building on these advances, our approach uses GIB to extract the most informative subgraph that captures ID patterns. This subgraph plays a crucial role in retaining in-distribution knowledge during continual adaptation, ensuring that the model maintains high performance on ID data while adapting to evolving OOD data. Unlike prior methods, which fail to retain ID-specific knowledge, our framework guarantees the continual relevance of ID parameters, providing superior performance across both ID and OOD samples in real-world scenarios.

## 3 PROPOSED APPROACH

### 3.1 PROBLEM FORMULATION

Let $P(\mathbf{x})$ be the distribution of the source training data $\{\mathbf{x}^{(i)}\}_{i=1}^{|\mathcal{S}|}$, i.e., $\mathbf{x}^{(i)} \sim P(\mathbf{x})$, and $\{\mathbf{x}^{(i)}, \mathbf{y}^{(i)}\}_{i=1}^{|\mathcal{S}|}$ be the labeled data with $|\mathcal{S}|$ pairs. We note that $\mathbf{x} = \{\boldsymbol{x}_1, \boldsymbol{x}_2, ..., \boldsymbol{x}_T\}$ is the historical poses of a person, and $\mathbf{y} = \{\boldsymbol{y}_1, \boldsymbol{y}_2, ..., \boldsymbol{y}_{\Delta T}\}$ is the corresponding future poses, with each frame $\boldsymbol{x}_t$ and $\boldsymbol{y}_t$ containing the 3D coordinates of the $J$ human joints (Dang et al., 2021; Li et al., 2022d; Wang et al., 2023). Given a base model $f_{\Theta^{(0)}}$ trained on the source domain $\{\mathbf{x}^{(i)}, \mathbf{y}^{(i)}\}_{i=1}^{|\mathcal{S}|}$, the distribution shift is inevitable in practice, where the model frequently encounter the out-of-distribution data $\mathbf{x} \sim Q(\mathbf{x})$, with $Q(\mathbf{x}) \neq P(\mathbf{x})$ (Su et al., 2023; Wang et al., 2020). Under this situation, the prediction of the base model $f_{\Theta^{(0)}}(\mathbf{x})$ will significantly degrade.

To this end, TTA aims to adapt the model to the target domain, and boost the performance of out-of-distribution data (Brahma & Rai, 2023; Cui et al., 2023b;a). Concretely, given a batch of test sequences $\{\mathbf{x}^{(i)}\}_{i=1}^{B}$, $\mathbf{x}^{(i)} \sim Q(\mathbf{x})$, TTA methods fine-tune the model $f_{\Theta^{(0)}}$ to achieve $f_{\Theta^*} \leftarrow f_{\Theta^{(0)}}$ to adapt to the target distribution $Q(\mathbf{x})$. One can achieve this by minimizing some self-supervised loss $\mathcal{L}_{\text{self}}$ on the target sequences (Liu et al., 2021b; Tomar et al., 2023; He et al., 2021), defined as:

$$\min_{\Theta^*} \mathcal{L}_{self}(\mathbf{x}; \Theta), \ \mathbf{x} \sim Q(\mathbf{x}). \tag{1}$$

We note that the existing TTA-based HPP models (Cui et al., 2023a;b) assume that the target domain is stationary, i.e., $Q(\mathbf{x}) = Q(\mathbf{x}_1) = Q(\mathbf{x}_2) = ... = Q(\mathbf{x}_n)$, and all target sample are drawn from out-of-distribution, i.e., $Q(\mathbf{x}) \subset Q_{\text{ood}}(\mathbf{x})$ and $Q_{\text{ood}} \neq P(\mathbf{x})$. This work breaks this assumption and proposes a novel paradigm, which includes the following distinctions:

- Owing to environmental changes and individual behavioral habits, the target domains will constantly change, i.e., $Q(\mathbf{x}) \neq Q(\mathbf{x}_1) \neq Q(\mathbf{x}_2) \neq ... \neq Q(\mathbf{x}_n)$, $n > 1$. It necessitates the model to continually adapt to the changing target domain.

- In HPP scenarios, the target domain $Q(\mathbf{x})$ contains both ID and OOD data, that is, $Q(\mathbf{x}) = Q_{\text{ood}}(\mathbf{x}) \cup Q_{\text{id}}(\mathbf{x})$, where $Q_{\text{ood}}(\mathbf{x}) \neq P(\mathbf{x})$ and $Q_{\text{id}}(\mathbf{x}) \approx P(\mathbf{x})$. Simply making adaptation to the OOD data may lead to significant performance degradation on ID test sequences.

Our proposed framework, IDKR, addresses these challenges through several key innovations: 1) We introduce an in-distribution (ID) informative subgraph learning method tailored for graph-like human skeleton structures, which extracts the most relevant subgraph with respect to the ID labels. 2) We employ a structural graph Fisher information regularization to quantify the importance of model parameters based on the informative subgraph. Higher values indicate ID-specific parameters, while lower values correspond to OOD-specific parameters. 3) We integrate this regularization into the self-supervised loss function to preserve ID-specific knowledge while allowing other parameters to adapt during the continual adaptation process.

### 3.2 IN-DISTRIBUTION INFORMATIVE SUBGRAPH LEARNING

Consider a skeleton sequence $\mathbf{x}$ represented as an undirected graph $\mathcal{G} = (\mathbf{x}, \mathcal{A})$, where $\mathbf{x} \in \mathbb{R}^{K \times d}$ denotes the joint features, with $K = \Delta T \times J$. The adjacency matrix $\mathcal{A} \in \{0, 1\}^{K \times K}$ encodes connections between parent-child joint pairs and between corresponding joints across adjacent frames. The graph label $\mathcal{Y}$ indicates whether the sequence is in-distribution (ID, $\mathcal{Y} = 0$) or out-of-distribution (OOD, $\mathcal{Y} = 1$). Building on recent advances in Graph Invariant Learning (GIL) (Chen et al., 2024b; Li et al., 2022c), which focus on extracting critical subgraph structures while ignoring non-essential elements, we propose a GIL-based method to detect OOD data by isolating subgraphs correlated with the ID label. Our approach identifies these invariant subgraphs, which capture essential ID-specific patterns across motion sequences. To achieve this, we introduce a graph manipulator $\mathcal{M} = (\mathcal{M}_{\mathbf{x}}, \mathcal{M}_{\mathcal{A}})$, where $\mathcal{M}_{\mathbf{x}} \in \mathbb{R}^{K \times d}$ and $\mathcal{M}_{\mathcal{A}} \in \mathbb{R}^{K \times K}$ serve as binary masks, enabling the extraction of informative subgraph structures. The resulting subgraph $\mathcal{Z} = \mathcal{G} \odot \mathcal{M} = (\mathcal{G} \odot \mathcal{M}_{\mathbf{x}}, \mathcal{G} \odot \mathcal{M}_{\mathcal{A}})$ encapsulates the necessary information for OOD detection, while irrelevant or detrimental features are removed (Mao et al., 2024).

GIB (Wu et al., 2020; Ding et al., 2024) further refines this process by compressing graph information, retaining only what is most valuable for distinguishing ID data. Inspired by GIB, our objective

Figure 1: Illustration of in-distribution informative subgraph learning. Given a graph $\mathcal{G}$, we learn a manipulator $\mathcal{M}$ that filters out the irrelevant elements and extracts the most informative subgraph $\mathcal{Z}$. This ID-informative subgraph stores the invariant information of $\mathcal{G}$ w.r.t. in-distribution knowledge. $g_\psi$ is graph representation learning model—GraphCL (You et al., 2020), and trained the source domain data annotated as ID labels. Moreover, although the training objective is $\mathcal{L}_{\text{GIB}}$ of the predicted label and actual one, at test time, the predicted label is not required, but the informative subgraph w.r.t. the ID label is used to identify the ID-specific parameters.

is to maximize the mutual information between the compressed subgraph $\mathcal{Z}$ and the ID label $\mathcal{Y}$, while minimizing the mutual information between $\mathcal{Z}$ and the original graph $\mathcal{G}$:

$$\max_{\mathcal{Z}} I(\mathcal{Z}, \mathcal{Y}) - \alpha I(\mathcal{Z}, \mathcal{G}), \tag{2}$$

where $\alpha = 0.3$ is a Lagrange multiplier balancing these two objectives. To efficiently solve this problem, we introduce a variational lower bound on mutual information, reformulating Eq. 2 as:

$$\mathcal{L}_{\text{GIB}} = \frac{1}{B} \sum_{i=1}^{B} \left[ -\log q(\mathcal{Y}_i | \mathcal{Z}_i) + \alpha D_{KL} \left( q(\mathcal{Z}_i | \mathcal{G}_i) || p(\mathcal{Z}_i) \right) \right],$$

$$\approx \mathcal{L}_{CE}(p(\mathcal{Y}_i | \mathcal{Z}_i), \mathcal{Y}_i) + \alpha D_{KL} \left( q(\mathcal{Z}_i | \mathcal{G}_i) || p(\mathcal{Z}_i) \right), \tag{3}$$

where $\mathcal{L}_{CE}$ represents cross-entropy loss, and $B$ is the batch size. After training, the graph manipulator $\mathcal{M}$ extracts ID-informative subgraphs $\mathcal{Z}$ from test graphs $\mathcal{G}$. The illustration of the in-distribution informative subgraph learning is shown in Figure 1. Detailed optimization steps from Eq. 2 to Eq. 3 are provided in Appendix-A.

For extracting the latent features of subgraphs, we utilize GraphCL (You et al., 2020), a well-established graph representation learning model. GraphCL is applied to the subgraph $\mathcal{Z}$, which is trained on source domain data (labeled as $\mathcal{Y} = 0$). Consistent with conventional OOD detection methods (Sun et al., 2019; Sehwag et al., 2021), our OOD detector employs a parametric Mahalanobis distance approach (Sun et al., 2022) for identifying OOD samples.

### 3.3 STRUCTURAL GRAPH FISHER REGULARIZATION

Let $f_{\Theta^{(0)}}$ denote the underlying human pose prediction (HPP) model, where $\Theta \in \mathbb{R}^P$ (Ma et al., 2022; Dang et al., 2021; Xu et al., 2023; Lou et al., 2024). During the continual test-time adaptation (CTTA) process, for a given sample $\mathbf{x}^{(t)}$ and the adapted model $f_{\Theta^{(t-1)}}$ from the previous step, the Structural Graph Fisher Information Matrix (SG-FIM) is designed to identify and retain key parameters for in-distribution (ID) sequences. This ensures the preservation of ID-specific knowledge while dynamically updating other parameters, leading to the model's adaptation $f_{\Theta^{(t)}} \leftarrow f_{\Theta^{(t-1)}}$.

SG-FIM extends the classical Fisher Information Matrix (FIM) (Vedantam et al., 2021; Tian & Lyu, 2024; Brahma & Rai, 2023) by incorporating the graph structure and features of the human skeleton. Unlike traditional FIM, which measures parameter sensitivity on regular data, SG-FIM is tailored to the characteristics of human motion data, as follows:

- **Graph topology consideration**: While FIM generally overlooks the data's inherent structure, SG-FIM explicitly accounts for the geometric topology of the human skeleton, reflecting the structural relationships between joints and their temporal dependencies.

- **Edge feature integration**: SG-FIM is able to analyze both node (joint) and edge (bone) features. The edges, representing the fixed bone lengths between parent and child joints, are particularly important in human motion, as these lengths remain invariant regardless of pose. This invariance provides a stable reference for motion prediction, improving the accuracy of SG-FIM in estimating ID-specific parameters. Traditional graph methods often ignore such constraints, making SG-FIM uniquely suited for skeletal data.

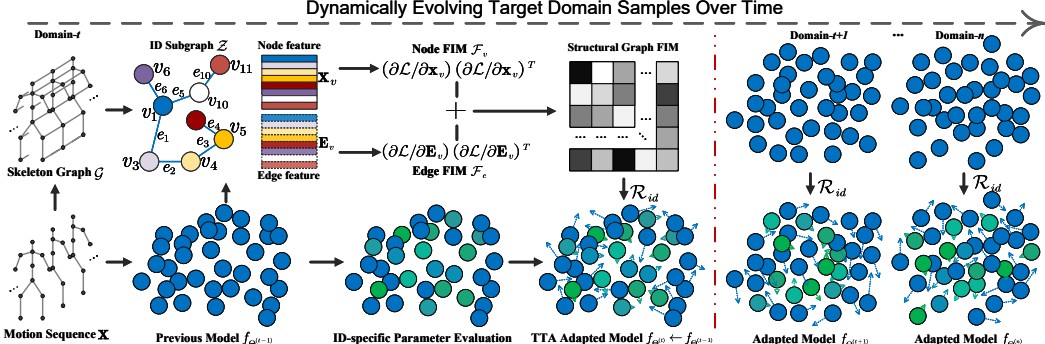

Figure 2: Overview of the proposed IDKR. Given a human skeleton sequence $\mathbf{x}$ at test time, the ID-informative subgraph $\mathcal{Z}$ is learned. For each joint $v$ and edge $e$ within $\mathcal{Z}$, we compute the node feature $\mathbf{x}_v$ and edge feature $\mathbf{E}_v$, followed by the calculation of gradients with respect to the model parameters $\Theta$. These gradients are then used to construct the Structural Graph Fisher Information Matrix (SG-FIM), which quantifies parameter importance based on both node and edge sensitivities. The diagonal elements of SG-FIM highlight the parameters most critical for retaining in-distribution (ID) knowledge—those with higher values (depicted in green) are prioritized for preservation during adaptation. This IDKR regularization mechanism ensures that ID-specific parameters (indicated by shorter arrows) are retained, while non-essential parameters (represented by longer arrows) are updated during the continual adaptation process.

- **Refinement of ID-specific parameters**: Instead of calculating parameter sensitivity with respect to the raw input data, SG-FIM evaluates sensitivity in relation to the ID-invariant subgraph. This refinement allows the model to more accurately pinpoint parameters critical for retaining ID knowledge during adaptation, thus ensuring robust performance in ID-targeted domains.

As illustrated in Figure 2, given that our objective is to identify ID-specific parameters, we apply the compressed subgraph $\mathcal{Z}$—obtained via the ID-informative subgraph learning process—instead of the original full skeleton graph $\mathcal{G}$. For each joint $v$ and edge $e$ in the subgraph $\mathcal{Z}$, we compute the gradient of the current model's output $f_{\Theta^{(t-1)}}$ with respect to the node and edge features:

$$\nabla_{v,\Theta} = \frac{\partial \mathcal{L}(\tilde{\mathbf{y}}, f_{\Theta^{(t-1)}}(\mathbf{x}))}{\partial \mathbf{x}_v}, \quad \nabla e, \Theta = \frac{\partial \mathcal{L}(\tilde{\mathbf{y}}, f_{\Theta^{(t-1)}}(\mathbf{x}))}{\partial \mathbf{E}_v}, \tag{4}$$

where $\tilde{\mathbf{y}} = f_{\Theta^{(0)}}(\mathbf{x})$ is the surrogate label generated by the source-trained model, $\mathbf{x}_v$ denotes the features of the $v$-th joint, and $\mathbf{E}_v$ represents the edge connecting the $v$-th joint to its adjacent node. The loss function $\mathcal{L}$ is computed as the L2 distance.

Next, we construct the FIM matrices with respect to the nodes and edges, denoted as $\mathcal{F}_v$ and $\mathcal{F}_e$, respectively, by computing the outer products of the gradients:

$$\mathcal{F}_v = \mathbb{E}_{\mathcal{Z}} \left[ \nabla_{v,\Theta} \nabla_{v,\Theta}^T \right], \quad \mathcal{F}_e = \mathbb{E}_{\mathcal{Z}} \left[ \nabla_{e,\Theta} \nabla_{e,\Theta}^T \right]. \tag{5}$$

The overall SG-FIM is obtained by aggregating the contributions of all nodes and edges in the subgraph $\mathcal{Z}$. For computational efficiency and enhanced interpretability, we assume the independence of model parameters and focus on the diagonal elements of SG-FIM to assess parameter importance:

$$\mathcal{F}_\Theta = \text{Diag} \left[ \sum_{v \in \mathcal{V}} \mathcal{F}_v(\Theta) + \sum_{e \in \mathcal{E}} \mathcal{F}_e(\Theta) \right]. \tag{6}$$

Here, $\mathcal{F}(\Theta) \in \mathbb{R}^P$ has the same dimensionality as the parameter vector $\Theta$. The diagonal elements capture the sensitivity of each parameter to changes in the ID data, indicating the parameter's contribution to the model's predictive performance on ID sequences.

By incorporating the fixed bone lengths and graph topology into the sensitivity analysis, SG-FIM provides a more accurate estimation of the parameters crucial for ID-targeted human pose prediction. The ability to leverage both joint and edge features makes it particularly effective for motion data, where spatial relationships between joints remain invariant across different poses. This method significantly enhances model robustness in continuously adapting to new environments while preserving ID knowledge. Given its flexibility and generalizability, consistent with (Cui et al., 2023b), we adopt siMPLE (Guo et al., 2023) as the base HPP model $f_\Theta$.

### 3.4 Continual Adaptation with ID knowledge retention

After obtaining the diagonal SG-FIM, we propose a novel weighted Fisher regularization termed In-Distribution Knowledge Retention (IDKR). This mechanism is specifically designed to mitigate catastrophic forgetting, particularly addressing the performance degradation on in-distribution (ID) motion sequences (Wang et al., 2024). By introducing a regularization term based on SG-FIM into the TTA optimization process, we ensure that the parameters critical for ID sequences are preserved during continual adaptation (Tian & Lyu, 2024; Saharia et al., 2022; Press et al., 2024).

The diagonal SG-FIM reflects the contribution of each parameter to the model's predictive ability on ID data: larger values indicate parameters that have a greater impact on ID performance and should be retained, while smaller values suggest minimal impact and thus should be updated (Sanyal et al., 2023; Sójka et al., 2023). During parameter updates from time step $t-1$ to $t$, we adjust the magnitude of each parameter update by weighting it with the corresponding IDKR regularization:

$$\mathcal{R}_{id}(\Theta^{(t-1)}, \Theta^{(0)}) = \sum_{i=1}^{P} \mathcal{F}(\theta_i^{(t-1)}) \|\theta_i^{(t-1)} - \theta_i^{(0)}\|^2, \tag{7}$$

where $\Theta^{(0)}$ represents the parameter set of the base model $f_{\Theta^{(0)}}$, and $\theta_i \subseteq \Theta$ denotes an individual parameter. The term $\mathcal{F}(\theta_i)$ is the $i$-th element of the diagonal SG-FIM matrix, indicating the importance of $\theta_i$ for ID sequences. To achieve continual adaptation with ID knowledge retention, we integrate $\mathcal{R}_{id}$ into the self-supervised loss $\mathcal{L}_{\text{self}}$ (from Eq. 2) as follows:

$$\min_{\Theta} \mathcal{L}_{self}(\mathbf{x}; \Theta^{(t-1)}) + \beta \mathcal{R}_{id}(\Theta^{(t-1)}, \Theta^{(0)}), \tag{8}$$

where $\beta = 0.2$ is a trade-off parameter, and $\mathcal{L}_{self}$ represents the self-supervised loss function.

Instead of knowledge distillation (Cui et al., 2023b) or auxiliary learning (Cui et al., 2023a) where the accuracy of pseudo-labels degrades as adaptation progresses, we employ self-supervised loss functions to adapt the model to incoming test data in an online fashion, using both spatial and temporal smoothness constraints to ensure stable predictions. Moreover, we notice that for time-series prediction task of HPP, not all frames contribute equally to the prediction; instead, more recent frames carry more relevant information to the prediction, while older ones are less relevant. This motivates us to introduce the time-weighted term to form the time-weighted spatial loss ($\mathcal{L}_{\text{TWS}}$) and time-weighted temporal loss ($\mathcal{L}_{\text{TWT}}$), which assign different weights to each frame based on its temporal distance to the current frame. The spatial loss $\mathcal{L}_{\text{TWS}}$ is grounded in the observation that the relative positions of adjacent joints and bone lengths remain consistent across poses:

$$\mathcal{L}_{\text{TWS}} = \frac{1}{T(J-1)} \sum_{t=1}^{T} \sum_{j=1}^{J} w_t \cdot |l_{t,j}^{obs} - l_{t,j}^{pred}|, \tag{9}$$

where $l_{t,j}^{obs}$ and $l_{t,j}^{pred}$ are the observed and predicted bone lengths of joint $j$ in frame $t$, respectively. The weight $w_t = \exp(\gamma t) / \sum_{t=1}^{T} \exp(\gamma t)$ is a time-weighted function, and $\gamma = 0.7$ is a tunable parameter that controls the rate of exponential decay.

The time-weighted temporal loss $\mathcal{L}_{\text{TWT}}$ is introduced to enforce coherence across consecutive frames, minimizing abrupt changes in the adjacent predicted frames:

$$\mathcal{L}_{\text{TWT}} = \frac{1}{T-1} \sum_{t=1}^{T} w_t \cdot \|\hat{\boldsymbol{y}}_{t+1} - \hat{\boldsymbol{y}}_t - (\boldsymbol{x}_{t+1} - \boldsymbol{x}_t)\|^2. \tag{10}$$

Both self-supervised terms are integrated into the TTA optimization process to enable continual adaptation with ID knowledge retention:

$$\Theta^{(t)} \leftarrow \Theta^{(t-1)} - \eta \nabla_{\Theta} \left[ \mathcal{L}_{\text{TWS}}(\mathbf{x}, \hat{\mathbf{y}}; \Theta^{(t-1)}) + \mathcal{L}_{\text{TWT}}(\hat{\mathbf{y}}; \Theta^{(t-1)}) + \beta \mathcal{R}_{id}(\Theta^{(t-1)}, \Theta^{(0)}) \right], \tag{11}$$

where $\eta = 0.001$ is the learning rate, and $\hat{\mathbf{y}}$ represents the predicted sequence from the model $f_{\Theta^{(t-1)}}$. The final prediction sequence is generated by the adapted model $f_{\Theta^{(t)}}$.

## 4 Experiments

### 4.1 Datasets and Evaluation Protocols

**Datasets:** IDKR framework is evaluated on three widely-used human pose prediction benchmarks: (1) CMU MoCap (cmu, 2003) is a representative dataset, including 8 action categories; (2) H3.6M

Table 1: **Setup-**N: evaluation of general predictive ability. We highlight the best results in bold, and the second best in underlined. † indicates that the results are from Xu et al. (2023), ‡ indicates the re-implementation, and others are from the original paper. To distinguish, we mark our method as IDKR* with a star. For the baselines without the PCK@150 metric, we re-statistic the results.

| | | H3.6M | | | | CMU Mocap | | | | GRAB | | | |
|---|---|---|---|---|---|---|---|---|---|---|---|---|---|
| | milliseconds | 80 | 160 | 400 | 1000 | 80 | 160 | 400 | 1000 | 200 | 400 | 600 | 1000 |
| MPJPE | LTD† | 12.7 | 26.1 | 63.5 | 114.3 | 9.9 | 18.0 | 41.0 | 81.9 | 38.3 | 68.7 | 101.6 | 197.3 |
| | PGBIG† | 10.3 | 22.7 | 58.5 | 110.3 | 8.2 | 15.4 | 37.3 | 76.7 | 30.1 | 53.9 | 92.2 | 157.2 |
| | SPGSN† | 10.4 | 22.3 | 58.3 | 109.6 | 8.3 | 14.8 | 37.0 | 77.8 | 27.4 | 50.6 | 91.3 | 144.5 |
| | siMLPe | 9.6 | 21.7 | 57.3 | 109.4 | 8.3 | 14.6 | 37.2 | 76.6 | 27.1 | 51.5 | 88.4 | 137.5 |
| | H/P-TTP‡ | 9.8 | 21.1 | 55.6 | 103.7 | 8.0 | 13.1 | 35.3 | 74.4 | 26.5 | 47.4 | 85.5 | 138.0 |
| | IDKR* | 8.9 | 18.6 | 54.0 | 100.4 | 7.7 | 11.1 | 33.1 | 70.5 | 22.5 | 44.4 | 81.4 | 131.7 |
| PCK@150 | LTD† | 79.9 | 77.3 | 70.4 | 66.0 | 84.2 | 81.5 | 77.2 | 75.2 | 81.8 | 77.3 | 71.3 | 62.9 |
| | PGBIG† | 88.5 | 84.2 | 77.3 | 69.6 | 88.8 | 83.2 | 78.0 | 77.0 | 84.3 | 82.2 | 75.8 | 66.4 |
| | SPGSN† | 87.8 | 84.7 | 80.1 | 71.2 | 88.4 | 85.1 | 77.9 | 76.4 | 87.1 | 80.4 | 77.0 | 67.8 |
| | siMLPe | 88.4 | 86.6 | 83.4 | 72.7 | 90.0 | 88.2 | 83.7 | 77.5 | 86.9 | 82.6 | 82.1 | 69.1 |
| | H/P-TTP‡ | 91.2 | 89.4 | 85.1 | 74.6 | 91.3 | 89.4 | 84.7 | 79.4 | 88.0 | 82.3 | 81.1 | 70.4 |
| | IDKR* | 92.3 | 90.2 | 87.0 | 77.1 | 93.5 | 91.0 | 86.8 | 81.7 | 88.0 | 84.7 | 83.3 | 72.5 |

(Ionescu et al., 2014) contains $\approx 3.6$M frames of 7 subjects performing 15 actions; (3) GRAB (Taheri et al., 2020) is newly-introduced with $\approx 1.6$M poses of 29 actions from 10 human subjects. Compared with H3.6M, the pose sequences in GRAB are more diverse and involve interaction with the physical world, making it a more challenging dataset. For all 3 datasets, each pose is specified by 3D coordinates of 17 joints, and normalized to $[-1, 1]$. All methods are implemented to predict the next 1 second, with the observed length of 1 second.

**Protocols:** (1) Mean Per Joint Position Error (MPJPE) (Lou et al., 2024; Ma et al., 2022) measures the average Euclidean distance between the predicted and ground-truth 3D joint positions; (2) Percentage of Correct Keypoints (PCK) (Habibie et al., 2019) computes the percentage of correctly predicted keypoints within a certain threshold of the ground-truth keypoints. Following the literature, we use the threshold of 150mm, termed as PCK@150.

## 4.2 EXPERIMENTAL SETUPS AND BASELINES

**Experimental Setups:** It contains 4 similar experimental setups, including *Setup-*N, *Setup-*C$^+$, *Setup-*S$^+$ and *Setup-*D$^+$, as in the previous literature (Cui et al., 2023b;a), along with 2 newly-designed experimental setups, i.e., *Setup-*C$^{+-}$ and *Setup-*S$^{+-}$, as follows:

(1) **Setup-**N: While IDKR aims to solve the domain-shift issue in HPP, the normal data split is also required for the generative prediction ability, which is termed as *Setup-*N ('N' means 'Normal'); (2) **Setup-**C$^+$ consider adapting the model to unseen motion categories, where 'C$^+$' denotes 'New Category'; Similarly, (3) **Setup-**S$^+$ is designed to evaluate the model's performance on unseen subjects; Considering that real deployment scenarios contain OOD and ID data, we design two new experimental setups: *Setup-*C$^{+-}$ and *Setup-*S$^{+-}$. (4) **Setup-**C$^{+-}$ differs from *Setup-*C$^+$ in that it assigns 10% of the source domain data from *Setup-*C$^+$ to the target domain; and similarly, (5) **Setup-**S$^{+-}$ assigns 10% of *Setup-*S$^+$ to its target test domain. Both *Setup-*C$^{+-}$ and *Setup-*S$^{+-}$ simulate a mix of ID and OOD data for subjects and categories in the target domain. (6) **Setup-**D$^+$: we also further introduce a challenging setup to make the model adapt to new dataset, where the source data is from H3.6M and the target data is from GRAB. Since the target domain is a new dataset, *Setup-*D$^+$ covers *Setup-*C$^{+-}$ and *Setup-*S$^{+-}$ and is more challenging.

**Baselines:** Five state-of-the-art approaches emerged in recent years are selected as the baselines, including 1) GCN-based LTD (Mao et al., 2019), PGBIG Ma et al. (2022), and SPGSN (Li et al., 2022d); 2) MLP-based siMLPe (Guo et al., 2023); 3) TTA-based H/P-TTP (Cui et al., 2023b).

## 4.3 RESULTS AND ANALYSIS

**General Predictive Ability Analysis:** While our IDKR mainly focuses on the scenario of distribution shift of HPP, due to the inherent diversity and flexibility of human motion, there may includes a certain degree of difference between the training and test data within the same dataset. Therefore, it is necessary to evaluate the general predictive ability of the proposed method on the three datasets

Table 2: *Setup*-C$^+$ and *Setup*-S$^+$: prediction evaluation for new categories/subjects.

| | Results on *Setup*-S$^+$ | | | | | | | | Results on *Setup*-C$^+$ | | | | | | | | | | | |
| | H3.6M | | | | GRAB | | | | H3.6M | | | | CMU Mocap | | | | GRAB | | | |
| milliseconds | 80 | 160 | 400 | 1000 | 200 | 400 | 600 | 1000 | 80 | 160 | 400 | 1000 | 80 | 160 | 400 | 1000 | 200 | 400 | 600 | 1000 |
|---|---|---|---|---|---|---|---|---|---|---|---|---|---|---|---|---|---|---|---|---|
| **MPJPE** | | | | | | | | | | | | | | | | | | | | |
| SPGSN† | 13.7 | 27.2 | 58.3 | 107.0 | 31.2 | 56.8 | 94.2 | 159.3 | 12.8 | 26.3 | 63.6 | 115.6 | 12.3 | 14.8 | 43.6 | 88.5 | 34.7 | 56.4 | 97.0 | 150.8 |
| siMLPe | 13.0 | 25.6 | 55.3 | 102.5 | 30.1 | 57.2 | 94.1 | 155.9 | 11.8 | 25.3 | 60.5 | 112.8 | 11.4 | 15.8 | 44.9 | 84.7 | 36.4 | 57.8 | 96.7 | 151.4 |
| H/P-TTP‡ | 12.5 | 24.7 | 56.4 | 102.8 | 28.6 | 52.3 | 88.9 | 135.5 | 12.1 | 24.7 | 53.6 | 102.5 | 9.8 | 13.4 | 41.0 | 77.9 | 31.1 | 51.2 | 89.4 | 140.2 |
| IDKR* | 12.6 | 22.3 | 52.3 | 97.1 | 26.3 | 49.1 | 83.9 | 128.5 | 10.0 | 21.3 | 50.5 | 99.7 | 9.1 | 13.1 | 38.8 | 74.6 | 30.8 | 49.2 | 85.0 | 135.2 |
| **PCK@150** | | | | | | | | | | | | | | | | | | | | |
| SPGSN † | 80.2 | 76.4 | 70.8 | 66.4 | 81.8 | 73.3 | 70.4 | 66.5 | 81.7 | 76.6 | 74.0 | 67.8 | 78.7 | 74.8 | 72.1 | 71.3 | 81.0 | 75.9 | 75.2 | 65.4 |
| siMLPe | 83.8 | 80.1 | 75.2 | 68.5 | 84.0 | 80.0 | 76.4 | 67.7 | 80.0 | 75.3 | 73.1 | 68.9 | 80.3 | 77.7 | 74.2 | 71.0 | 80.2 | 77.5 | 73.6 | 67.8 |
| H/P-TTP‡ | 85.3 | 80.5 | 74.3 | 70.9 | 83.5 | 79.4 | 76.8 | 69.6 | 86.3 | 79.0 | 75.1 | 73.3 | 90.0 | 83.6 | 81.4 | 75.3 | 84.4 | 81.3 | 78.7 | 70.4 |
| IDKR* | 87.5 | 83.5 | 78.0 | 74.5 | 85.1 | 83.0 | 79.3 | 73.1 | 89.0 | 83.2 | 79.7 | 75.2 | 92.1 | 85.8 | 83.0 | 78.7 | 86.4 | 83.2 | 81.7 | 72.6 |

Table 3: *Setup*-C$^{+-}$ and *Setup*-S$^{+-}$: prediction evaluation for a hybridization of ID and OOD data.

| | Results on *Setup*-S$^{+-}$ | | | | | | | | Results on *Setup*-C$^{+-}$ | | | | | | | | | | | |
| | H3.6M | | | | GRAB | | | | H3.6M | | | | CMU Mocap | | | | GRAB | | | |
| milliseconds | 80 | 160 | 400 | 1000 | 200 | 400 | 600 | 1000 | 80 | 160 | 400 | 1000 | 80 | 160 | 400 | 1000 | 200 | 400 | 600 | 1000 |
|---|---|---|---|---|---|---|---|---|---|---|---|---|---|---|---|---|---|---|---|---|
| **MPJPE** | | | | | | | | | | | | | | | | | | | | |
| SPGSN† | 14.9 | 28.3 | 61.0 | 109.7 | 32.4 | 57.3 | 96.0 | 162.1 | 13.4 | 26.2 | 65.1 | 117.0 | 12.5 | 15.6 | 45.2 | 90.4 | 35.1 | 57.7 | 99.4 | 153.2 |
| siMLPe | 12.4 | 26.7 | 62.8 | 114.3 | 31.4 | 58.6 | 95.0 | 156.7 | 13.3 | 27.1 | 56.8 | 105.7 | 11.6 | 16.1 | 45.7 | 85.2 | 38.5 | 59.5 | 98.4 | 155.0 |
| H/P-TTP‡ | 12.9 | 25.5 | 58.0 | 104.2 | 29.7 | 53.0 | 89.7 | 136.2 | 12.5 | 25.3 | 54.7 | 104.0 | 10.1 | 14.7 | 42.5 | 79.5 | 32.0 | 53.0 | 91.4 | 143.1 |
| IDKR* | 12.7 | 22.9 | 52.7 | 99.0 | 27.1 | 49.3 | 85.2 | 130.5 | 10.3 | 22.5 | 52.1 | 101.4 | 9.4 | 13.7 | 40.2 | 76.0 | 31.5 | 50.3 | 88.0 | 137.5 |
| **PCK@150** | | | | | | | | | | | | | | | | | | | | |
| SPGSN † | 81.4 | 77.6 | 72.1 | 68.8 | 82.4 | 74.5 | 72.7 | 68.3 | 82.4 | 77.9 | 75.1 | 69.2 | 79.3 | 76.0 | 74.2 | 74.7 | 81.5 | 76.7 | 77.0 | 67.5 |
| siMLPe | 82.0 | 78.7 | 73.9 | 67.2 | 83.5 | 78.9 | 75.0 | 66.0 | 79.3 | 73.9 | 72.7 | 68.2 | 79.7 | 76.4 | 73.1 | 70.2 | 79.4 | 75.8 | 72.9 | 67.0 |
| H/P-TTP‡ | 83.7 | 80.0 | 73.1 | 68.7 | 82.8 | 78.0 | 75.2 | 68.4 | 85.6 | 78.3 | 73.3 | 71.7 | 89.7 | 82.9 | 80.8 | 74.6 | 84.0 | 80.4 | 77.5 | 68.8 |
| IDKR* | 87.1 | 82.8 | 76.9 | 72.8 | 84.7 | 81.4 | 77.7 | 71.9 | 87.9 | 81.0 | 77.4 | 72.8 | 91.3 | 83.3 | 82.2 | 76.7 | 85.2 | 82.1 | 80.9 | 71.2 |

using the common data split, i.e., *Setup*-N. The results are shown in Table 1, where two metric, i.e., MPJPE [mm] and PCK@150 [%], are used to evaluate the performance of 6 baselines across different time intervals. From the results,, compared with siMLPe, IDKR achieves better performance across all datasets and time intervals, with a reduction of 1.5%, 2.1%, and 1.8% in MPJPE on H3.6M, CMU Mocap, and GRAB, respectively. It evidences that the common data split exists distribution shift with varying degrees, which is the main reason for the superiority of TTA-based H/P-TTP and our IDKR. Moreover, IDKR achieves the best performance against other baselines, which demonstrates the effectiveness of our method in handling the distribution shift in HPP.

**Predictive Ability Analysis of Unseen Subjects/Categories:** Next, referring to *Setup*-C$^+$ and *Setup*-S$^+$, we evaluate the performance for unseen subjects and action categories. This experiment simulates the real-world scenario where the new human subjects or action categories are inevitable. The model is expected to adapt to a new subject or category during the test phase, and trained on the other domains. We note that due to the significant performance on *Setup*-N, we only compare our IDKR with siMLPe, SPGSN, and H/P-TTP. Table 2 reports the average results of different adaptation for all sequences of each subject and category. From the results, we observe that either for *Setup*-C$^+$ or *Setup*-S$^+$, our IDKR performs well, and achieves the best performance on almost time intervals. It indicates that IDKR is able to calibrate the domain shift in HPP imposed by new subjects, motion patterns, and even novel action categories through continual TTA with in-distribution knowledge retention. It also evidences its potential to be applied in various real-world scenarios.

**Predictive Ability Analysis for a mix of ID and OOD data:** We note that in the real-world deployment scenario, the target domain is unknown in advance, typically containing both ID and OOD data, where the former distribution is similar to the source domain, and the latter is different. Considering ths mixture of properties, TTA-based methods require to not only adapt to the OOD test sequence, but also maintain the predictive ability for the ID sequences. This compatibility of ID and OOD performance is not considered in the standard or TTA-based HPP systems. To investigate the performance of this scenario, the proposed IDKR and the comparison baselines are evaluated on *Setup*-C$^{+-}$ and *Setup*-S$^{+-}$, where the base source model is trained on 90% of the training data, and

Table 4: *Setup*-D$^+$: prediction evaluation for new dataset (trained on H3.6M, adaptation to GRAB).

| | | A1 passing | | | | A2 eating | | | | A3 drinking | | | | A4 lifting | | | | A5 squeeze | | | |
|---|---|---|---|---|---|---|---|---|---|---|---|---|---|---|---|---|---|---|---|---|---|
| milliseconds | | 200 | 400 | 800 | 1000 | 200 | 400 | 800 | 1000 | 200 | 400 | 800 | 1000 | 200 | 400 | 800 | 1000 | 200 | 400 | 800 | 1000 |
| | SPGSN† | 43.7 | 75.6 | 110.0 | 146.7 | 35.6 | 75.3 | 122.8 | 171.4 | 33.4 | 45.7 | 92.2 | 153.5 | 37.1 | 60.3 | 115.7 | 158.4 | 23.5 | 30.7 | 55.4 | 101.3 |
| MPJPE | siMLPe | 40.2 | 69.7 | 109.2 | 140.5 | 34.5 | 70.4 | 118.5 | 172.1 | 34.7 | 46.8 | 90.4 | 147.9 | 37.8 | 67.3 | 119.4 | 162.2 | 22.1 | 33.2 | 57.3 | 103.6 |
| | H/P-TTP‡ | 30.1 | 45.4 | 89.8 | 121.4 | 29.7 | 53.1 | 98.7 | 152.4 | 27.7 | 39.6 | 76.3 | 133.6 | 31.0 | 44.7 | 91.2 | 138.5 | 18.0 | 29.5 | 50.2 | 96.3 |
| | IDKR* | **26.5** | **42.4** | **83.5** | **117.1** | **27.0** | **47.4** | **95.3** | **146.9** | **25.8** | **36.7** | **71.6** | **127.9** | **27.1** | **41.4** | **86.2** | **132.0** | **16.4** | **25.6** | **47.5** | **91.7** |
| | SPGSN† | 63.2 | 56.8 | 55.0 | 51.4 | 58.8 | 53.4 | 51.0 | 49.6 | 57.8 | 56.0 | 50.7 | 49.7 | 57.3 | 51.4 | 47.0 | 45.6 | 72.2 | 67.8 | 64.6 | 62.7 |
| PCK@150 | siMLPe | 62.3 | 57.7 | 54.3 | 50.3 | 61.2 | 55.3 | 53.3 | 51.6 | 58.3 | 55.7 | 52.4 | 50.4 | 56.3 | 50.5 | 46.9 | 44.3 | 73.5 | 68.9 | 65.2 | 63.5 |
| | H/P-TTP‡ | 67.2 | 63.5 | 62.3 | 57.8 | 65.2 | 57.4 | 56.6 | 55.2 | 63.3 | 60.7 | 57.7 | 55.8 | 60.0 | 55.8 | 49.7 | 47.7 | **80.2** | 75.3 | 71.4 | 67.7 |
| | IDKR* | **70.3** | **65.7** | **63.2** | **61.0** | **66.7** | **60.5** | **59.0** | **57.2** | **65.1** | **62.3** | **60.6** | **58.4** | **63.1** | **58.4** | **53.0** | **51.0** | 78.4 | **76.7** | **73.4** | **70.5** |

the remaining 10% is merged into the original target domain for adaptation. The results are reported in Table 3, and it is observed that our IDKR obtains the better prediction results compared to the other baselines. Moreover, even for TTA-based H/P-TTP, IDKR achieves a significant improvement in MPJPE and PCK@150, attributed to the IDKR's ability to retain the ID-specific knowledge. Therefore, during long-term TTA process, our approach exhibits a superior adaptation ability of OOD data, while bringing better performance on ID data, which is crucial for real-world deployment scenarios. The separate results of the specific ID and OOD data are presented in Appendix-C.

**Predictive Ability Analysis for New Dataset:** Finally, we evaluate the performance of the proposed IDKR on *Setup*-D$^+$, where the source domain is H3.6M and the target domain is GRAB. Note that the acquisition environments of H3.6M and GRAB are completely different, where the former is typically used to the standard action analysis tasks, and the latter is captured in object manipulation scenarios. Moreover, the human subjects and action categories in the two datasets are different. Therefore, *Setup*-D$^+$ can be considered as a composite of the previous experimental setups, i.e., *Setup*-C$^{+-}$ and *Setup*-S$^{+-}$, and is more challenging. For simplicity, we categorize the sub-actions of the GRAB dataset into 5 cases according to the similarity of the action types, including A1 passing, A2 eating, A3 drinking, A4 lifting, and A5 squeeze. The results are presented in Table 4, statistic the average results across all sequences of each sub-action. We observe that IDKR achieve the best performance, which is mainly attributed to the fact that the new GRAB dataset, although collected from a different condition, contains both similar ID distribution as H3.6M, as well as a differential OOD distribution. Our IDKR is able to adapt to the specific properties of the OOD sequences, while also perform well well for the ID distribution during continual TTA procedure.

**Visualization:** In addition to the numerical evaluation, we also visualize the prediction of our IDKR and the state-of-the-art H/P-TTP of the 'airplane-fly' activity under *Setup*-D$^+$. As shown in Figure 4 of Appendix-D, the prediction of our IDKR is more accurate, and closer to the ground truth.

**Ablation Studies:** We also conduct ablation studies to investigate the effectiveness of the proposed IDKR. Please refer to the Appendix-F for the detailed analysis.

## 5 CONCLUSION

In this work, we address a more realistic TTA scenario for human pose prediction, wherein both in-distribution and out-of-distribution motion sequences are present in the target deployment domain. To tackle this challenge, we propose a novel continual TTA method incorporating an in-distribution knowledge retention mechanism. Our approach utilizes the Graph Information Bottleneck framework to compress the most informative subgraph relative to in-distribution for any target skeleton sequence, which facilitates filtering out irrelevant elements or structures. Using this subgraph, we calculate a structural graph fisher information matrix to identify parameters that significantly contribute to the prediction of in-distribution sequences. It is then constructed an in-distribution knowledge retention regularization, which is integrated into the TTA optimization process to control the preservation of in-distribution parameters during continual TTA. Extensive experiments demonstrate that the proposed IDKR outperforms state-of-the-art methods across various real-world experimental setups, thereby evidencing its practicality and effectiveness.

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

# Appendix

## A  OPTIMIZATION OF MINIMIZING THE MUTUAL INFORMATION

In this sub-section, we provide the detailed optimization process of the graph information bottleneck (GIB) as in Eq. 2. Recently, graph information bottleneck (GIB) has been integrated into GIL theory, where the bottleneck is engineered to compress the original graph information, preserving what is useful for detecting the ID. Inspired by this, given a graph $\mathcal{G}$ and its label $\mathcal{Y}$, our objective is to learn a compressed subgraph $\mathcal{Z}$ that maximizes the mutual information with the ID label $\mathcal{Y}$ while containing minimal mutual information with the original graph $\mathcal{G}$. For the sake of convenience, we re-write the GIB objective in Eq. 2 as follows:

$$\max_{\mathcal{Z}} I(\mathcal{Z}, \mathcal{Y}) - \alpha I(\mathcal{Z}, \mathcal{G}). \tag{12}$$

$\alpha$ is the Lagrange multiplier. Once the invariant subgraph $\mathcal{Z}$ is obtained, it can be fed into a GNN model $g_\psi$ to extract the latent node and edge feature.

It is difficult to directly optimize the mutual information $I(\mathcal{Z}; \mathcal{Y})$ due to the intractability of the marginal distribution $p(\mathcal{Z})$ and $p(\mathcal{G})$ (Shen et al., 2021; Higgins et al., 2016; Covert et al., 2023). Therefore, for the sake of convenience, we propose to maximize the lower bound of the mutual information to simplify the optimization process (Cha et al., 2022; Li et al., 2022a). To be specific, we introduce a variational distribution $q(\mathcal{Z}|\mathcal{G})$, controlled by the learnable parameter $\phi$, which given the graph $\mathcal{G}$, denotes the distribution of the compressed subgraph $\mathcal{Z}$. Then, our objective is to seek a distribution $q(\mathcal{Z}|\mathcal{G})$ that maximizes $I(\mathcal{Z}; \mathcal{Y})$ while minimizing $I(\mathcal{Z}; \mathcal{G})$. For this purpose, we introduce a variational lower bound of the mutual information, which can be formulated as:

$$I(\mathcal{Z}; \mathcal{Y}) \geq \mathbb{E}_{q(\mathcal{Z}|\mathcal{G})} \left[ \log p(\mathcal{Y}|\mathcal{Z}) - D_{KL} \left( q(\mathcal{Z}|\mathcal{G}) || p(\mathcal{Z}) \right) \right], \tag{13}$$

where $D_{KL}(q(\mathcal{Z}|\mathcal{G})||p(\mathcal{Z}))$ is the Kullback-Leibler divergence between the variational distribution $q(\mathcal{Z}|\mathcal{G})$ and marginal distribution $p(\mathcal{Z})$. Then, the objective of GIB in Eq. 12 can be rewritten as:

$$\max_{\psi, \phi} \mathbb{E}_{q(\mathcal{Z}|\mathcal{G}; \phi)} \left[ \log p(\mathcal{Y}|\mathcal{Z}; \psi) - \alpha D_{KL} \left( q(\mathcal{Z}|\mathcal{G}; \phi) || p(\mathcal{Z}) \right) \right], \tag{14}$$

where $\psi$ is the learnable parameter of the GNN model $g_\psi$ for $p(\mathcal{Y}|\mathcal{Z})$. For simplicity, $\phi$ is implemented as the gaussian distribution. Eq. 14 can be further optimized by the following loss function:

$$\mathcal{L}_{\text{GIB}} = \frac{1}{B} \sum_{i=1}^{B} \left[ -\log q(\mathcal{Y}_i|\mathcal{Z}_i) + \alpha D_{KL} \left( q(\mathcal{Z}_i|\mathcal{G}_i) || p(\mathcal{Z}_i) \right) \right],$$
$$\approx \mathcal{L}_{CE}(p(\mathcal{Y}_i|\mathcal{Z}_i), \mathcal{Y}_i) + \alpha D_{KL} \left( q(\mathcal{Z}_i|\mathcal{G}_i) || p(\mathcal{Z}_i) \right), \tag{15}$$

where $\mathcal{L}_{CE}$ is the cross-entropy loss, and $B$ is the batch size. Once the training is completed, for a test human skeleton graph $\mathcal{G}$, the graph manipulator $\mathcal{M}$ is used to obtain the ID-informative subgraph $\mathcal{Z}$.

To achieve graph OOD detection, $g_\psi$ is implemented as GraphCL (You et al., 2020) which is a representative graph representation learning model, and can be used to extract the latent feature of any graph structure. Note that the model follows the open-source weights and is trained on our source domain data, where the training data only contains the source domain, labeled as $\mathcal{Y} = 0$. Consistent with typical methods (Sun et al., 2019; Sehwag et al., 2021), the OOD detector D is designed as parametric approach Mahalanobis distance (Sun et al., 2022). Eq. 15 is able to obtain a OOD score/label and the compressed subgraph $\mathcal{Z}$; however, the latter is only used for the structural graph Fisher information matrix calculation in the IDKR framework.

## B  PROOF OF CONCEPT FOR HYBRIDIZATION OF ID AND OOD DATA IN HUMAN POSE PREDICTION TASK

Through the experimental results presented in the main manuscript, we have demonstrated the effectiveness of the proposed IDKR framework in addressing the distribution shift challenges in human pose prediction (HPP). Specifically, IDKR excels in scenarios involving a mixture of ID and OOD

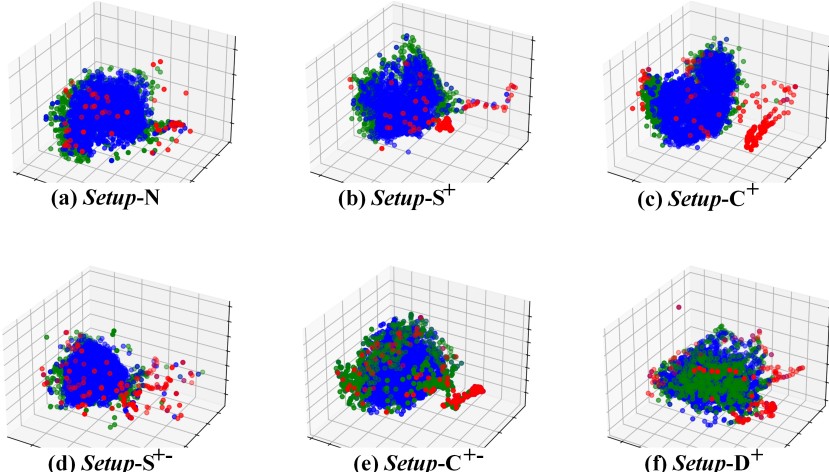

(a) *Setup*-N       (b) *Setup*-S$^+$       (c) *Setup*-C$^+$

(d) *Setup*-S$^{+-}$       (e) *Setup*-C$^{+-}$       (f) *Setup*-D$^+$

Figure 3: t-SNE visualization of the source domain (blue dots) and target domain of various experimental setups (c.f. Section 4.2). In each subfigure, 8192 source points and 1024 target points are uniformly sampled from the training and test set in different experiment setups. We note that the target domain includes both ID (green) and OOD (red) samples with distinct distributions. It needs to adapt to the OOD sequences while retaining the relevant knowledge to ensure optimal performance on target ID data in continual adaptation process.

data. This appendix provides a proof of concept to highlight the presence of such hybridization of ID and OOD data in HPP tasks. Our motivation is to confirm that real-world human motion prediction tasks indeed face a combination of ID and OOD samples alongside continuously evolving domain shifts, thereby justifying the development of IDKR.

Recent research has attempted to tackle domain shifts in HPP tasks using TTA, assuming a static target domain distribution that differs entirely from the source domain. However, this assumption deviates from practical scenarios where target domain distributions are dynamic and often comprise a combination of ID and OOD data. In real-world deployment, it is common to encounter data resembling the source domain (ID data) as well as data that deviates from it (OOD data). We define this as the hybrid ID and OOD human pose prediction problem, which is a novel and more realistic problem setting for HPP tasks. To this end, we propose leveraging an In-Distribution Knowledge Retention mechanism to preserve ID-specific knowledge during continual TTA.

To validate our motivation, we present a proof of concept illustrating the presence of hybrid ID and OOD data in the HPP task. Specifically, we utilize 6 experimental setups (c.f. Section 4.2), namely *Setup*-N, *Setup*-C$^+$, *Setup*-S$^+$, *Setup*-C$^{+-}$, *Setup*-S$^{+-}$, and *Setup*-D$^+$. These setups encompass standard HPP data splits, scenarios with a mixture of ID and OOD data, and even new dataset conditions. For each experimental setup, we uniformly sample 8192 source data points and 1024 target data points from both the source and target domains. We then apply t-SNE to visualize these data points in a three-dimensional space, where each point represents a skeleton sequence. Source domain data points are depicted in blue, while target domain data points are shown in green and red, representing ID and OOD data, respectively. The distinction between ID and OOD data is determined using the OOD detection method described in the main manuscript.

The visualization results, as illustrated in Figure 3, clearly show varying degrees of hybridization between ID and OOD data across all experimental setups. Even in the relatively less complex *Setup*-N (Figure 3(a)), we observe the presence of both ID and OOD data. On the other hand, the *Setup*-D$^+$, as in Figure 3(f), which involves adapting the base model to a new data acquisition environment, exhibits the most significant hybridization of ID and OOD data. This setup reflects a realistic application scenario where a model trained on one dataset is applied to another, a common issue in practical deployment scenarios.

The proof of concept substantiates the following conclusions: 1) Hybridization of ID and OOD data is inherent in current HPP tasks. 2) To be effectively deployed, a motion prediction model must be

capable of adapting to both ID and OOD data simultaneously. This indicates that the motivation behind our IDKR method is sound and has practical relevance in real-world applications.

## C  SEPARATE RESULTS OF ID AND OOD DATA FROM TARGET DOMAIN

In the main manuscript, we construct two new experimental setups, *Setup*-$C^{+-}$ and *Setup*-$S^{+-}$, by randomly selecting 10% of the data from *Setup*-$C^+$ and *Setup*-$S^+$ as target domain data to simulate a scenario involving mixed ID and OOD data. The results for these two setups are presented in Table 3, where the ID and OOD data from the target domain are aggregated for evaluation. However, since this experimental setup is manually constructed, the ID and OOD attributes of each target domain sample are known. Consequently, it is feasible to separate and analyze the performance of the model on ID and OOD data individually. Such an analysis is crucial to understanding whether our proposed model maintains its predictive capability for ID samples while adapting to OOD samples, as compared to baseline methods.

Table 5 and Table 6 report the separate performance results for ID and OOD data in the *Setup*-$C^{+-}$ and *Setup*-$S^{+-}$ configurations. From the separate results, our IDKR consistently achieves the best predictive results for both ID and OOD data. For the OOD data, our proposed method performs slightly better than the H/P-TTP method across both datasets. Notably, for the ID samples in the target domain, our method demonstrates significant improvements at all time scales. This indicates that our approach effectively retains the predictive capability for ID data while adapting to OOD data, which is a crucial ability for HPP system in real-world deployment scenarios.

Table 5: *Setup*-$C^{+-}$ and *Setup*-$S^{+-}$: Prediction evaluation for a mix of ID and OOD data.

| | | Results on *Setup*-$S^{+-}$ | | | | | | | | | | | | | | | |
| | | ID (H3.6M) | | | | OOD (H3.6M) | | | | ID (GRAB) | | | | OOD (GRAB) | | | |
| milliseconds | | 80 | 160 | 400 | 1000 | 80 | 160 | 400 | 1000 | 200 | 400 | 600 | 1000 | 80 | 160 | 400 | 1000 |
| MPJPE | SPGSN† | 13.2 | 27.1 | 58.5 | 105.4 | 17.4 | 29.3 | 64.1 | 109.0 | 30.1 | 55.5 | 94.3 | 158.7 | 33.4 | 58.2 | 98.7 | 163.2 |
| | siMLPe | 11.7 | 24.6 | 60.3 | 112.7 | 14.3 | 27.1 | 63.2 | 114.0 | 30.5 | 56.9 | 93.1 | 153.5 | 31.4 | 57.9 | 96.3 | 158.4 |
| | H/P-TTP‡ | 11.8 | 24.6 | 56.2 | 103.4 | 12.3 | 25.6 | 57.1 | 105.0 | 29.7 | 53.0 | 89.7 | 136.2 | 30.7 | 55.1 | 91.5 | 138.6 |
| | IDKR* | 12.1 | 21.4 | 51.9 | 98.3 | 12.7 | 22.6 | 52.7 | 99.8 | 27.1 | 49.3 | 85.2 | 130.5 | 28.4 | 51.0 | 86.4 | 132.0 |

Table 6: *Setup*-$C^{+-}$ and *Setup*-$S^{+-}$: Prediction evaluation for a mix of ID and OOD data.

| | | Results on *Setup*-$C^{+-}$ | | | | | | | | | | | | | | | |
| | | ID (H3.6M) | | | | OOD (H3.6M) | | | | ID (GRAB) | | | | OOD (GRAB) | | | |
| milliseconds | | 80 | 160 | 400 | 1000 | 80 | 160 | 400 | 1000 | 200 | 400 | 600 | 1000 | 80 | 160 | 400 | 1000 |
| MPJPE | SPGSN† | 13.1 | 25.6 | 64.7 | 115.9 | 13.6 | 26.8 | 66.8 | 117.0 | 34.3 | 55.8 | 97.6 | 151.4 | 35.5 | 57.1 | 99.4 | 154.2 |
| | siMLPe | 13.0 | 26.5 | 57.1 | 103.9 | 13.5 | 27.8 | 59.3 | 105.1 | 36.9 | 57.5 | 97.2 | 154.4 | 37.4 | 59.4 | 99.7 | 157.3 |
| | H/P-TTP‡ | 12.1 | 24.5 | 53.0 | 103.4 | 12.6 | 26.1 | 55.7 | 105.7 | 31.1 | 52.2 | 89.9 | 141.3 | 33.2 | 54.7 | 83.1 | 144.0 |
| | IDKR* | 10.0 | 21.7 | 51.5 | 100.6 | 10.7 | 22.5 | 53.0 | 98.9 | 30.8 | 49.7 | 87.6 | 135.9 | 32.6 | 52.4 | 90.5 | 138.7 |

## D  VISUALIZATION OF PREDICTION RESULTS

In this section, we present the visualization of the prediction results for the 'lift-on' and 'fly-on' activities from the GRAB dataset. To thoroughly demonstrate the predictions of different methods, all visualizations are conducted under the *Setup*-$D^+$ experimental configuration. Instead of visualizing all baseline methods, we focus on comparing the predictions of the H/P-TTP method (Cui et al., 2023a) and our IDKR method, as H/P-TTP is the most representative HPP approach in the comparison, utilizing test-time adaptation.

As illustrated in Figure 4, the top sub-figure display the prediction results for the 'lift-on' activity, while the bottom sub-figure show the 'airplane-fly' activity. In each sub-figure, the upper illustrates the results of the H/P-TTP method, and the lower shows the results of our IDKR method. For clarity, the predictions within 1 second are divided into 2 time intervals: 0-500ms and 500-1000ms,

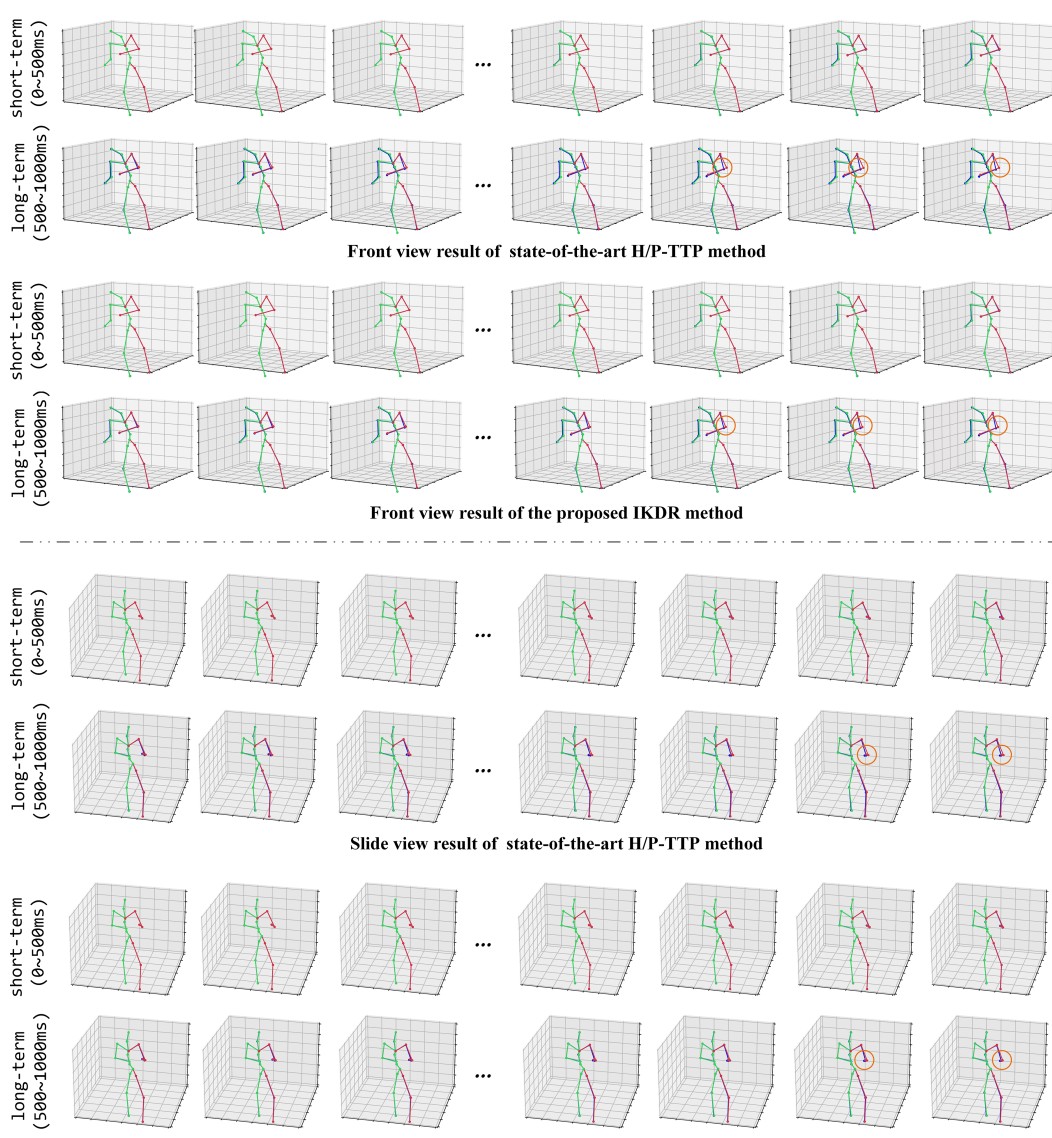

Figure 4: Visualization of the prediction results for the 'airplane-fly' activities from the GRAB dataset under the *Setup*-D$^+$ experimental configuration. We highlight the contrasting segments using orange circles, indicating the significant differences between the predictions made by H/P-TTP and our IDKR method. From the result, it is evident that our IDKR method generates more accurate predictions, especially in long-term predictions.

corresponding to short-term and long-term predictions. In each animation frame, the deep blue skeleton represents the ground truth, while the green-red skeleton denotes the predicted results. The orange circles highlight segments where the differences between the two methods are pronounced.

From the visualization results, it is evident that the predictions made by our IDKR method are closer to the ground truth, especially in long-term predictions. This further demonstrates the superiority of our IDKR method in generating accurate prediction outcomes.

## E  COMPARISON WITH OTHER TTA METHODS FROM NON-HPP TASKS

To further investigate the impact of our IDKR method, we compare it with other TTA methods that do not focus on HPP tasks. Concretely, 2 standard TTA methods, i.e., TENT and PETAL, and 1 con-

tinual TTA method, i.e., CoTTA, are selected for comparison: 1) TENT commences with a source model and exclusively updates the trainable BN parameters; 2) PETAL proposes a probabilistic CTTA, which regularizes the model updates at inference time to prevent model drift; 3) CoTTA is a continual TTA methods that utilize teacher-student learning to adapt model to the target domain.

We note that all experiments are conducted on the experimental *Setup*-$D^+$, as described in Section 4.2, where the source domain is H3.6M and the target domain is GRAB. Other experimental configurations are similar to the experimental setup in the main paper. As shown in Table 7, the performance of the proposed IDKR method achieves the best performance in terms of MPJPE, which evidences our method's effectiveness in addressing the distribution shift challenges and the hybridization of ID and OOD data in HPP tasks.

Table 7: Evaluation of the proposed IDKR method compared with other TTA or CTTA methods from non-HPP tasks on the *Setup*-$D^+$ experimental configuration.

|  | 200ms | 400ms | 800ms | 1000ms |
|---|---|---|---|---|
| TENT (Wang et al., 2020) | 31.4 | 42.4 | 83.2 | 137.3 |
| PETAL (Brahma & Rai, 2023) | 31.0 | 40.5 | 79.4 | 133.4 |
| CoTTA (Wang et al., 2022) | 26.4 | 40.7 | 80.1 | 129.5 |
| IDKR* (Ours) | 24.6 | 38.7 | 76.8 | 123.1 |

## F    ABLATION STUDIES

To investigate the influence of different components in our IDKR method, we conduct the ablation studies under the experimental setup *Setup*-$D^+$. Notably, only the MPJPE is used as the evaluation metric, and the other metrics are not used in the ablation studies.

**Effect of IDKR Regularization:** The IDKR regularization is designed to preserve in-distribution knowledge during continual test-time adaptation, thereby mitigating catastrophic forgetting of ID-specific parameters. As shown in Table 8 the model performance our proposed method (w/ IDKR regularization) significantly surpasses that w/o IDKR across. This demonstrates the effectiveness of the regularization in maintaining the predictive capability for ID samples, while simultaneously adapting to out-of-distribution (OOD) data. This improvement is attributed to the IDKR's ability to selectively update parameters based on the structural graph Fisher information matrix (SG-FIM), which accurately identifies and retains the essential parameters for ID data. In contrast, the absence of IDKR leads to a uniform update of all model parameters, causing significant performance degradation on ID sequences, especially in scenarios with mixed ID and OOD data.

**ID-informative Subgraph v.s. ID Label for Fisher Information Matrix:** This ablation study evaluates the effectiveness of using the ID-informative subgraph versus the direct use of ID labels in computing the Structural Graph Fisher Information Matrix (SG-FIM). As shown in Table 9, the performance of the model using the ID-informative subgraph significantly outperforms that of the model using ID labels. Specifically, the use of ID-informative subgraphs yields a lower MPJPE, indicating a more accurate capture of ID-specific parameters and improved model performance.

The ID-informative subgraph, derived from the GIL framework, effectively compresses the skeleton graph to retain only the most relevant substructures related to ID sequences. This selective representation allows the SG-FIM to focus on parameters that are crucial for ID data, enhancing the model's ability to preserve ID knowledge during adaptation. In contrast, directly using ID labels for SG-FIM computation leads to a less precise identification of ID-specific parameters, as the model cannot leverage the structural dependencies within the skeleton sequences. This results in suboptimal parameter updates and a notable increase in prediction error. This finding underscores the importance of structural representation learning in enhancing the effectiveness of continual TTA frameworks, particularly in complex scenarios with mixed ID and OOD data.

**Node Feature v.s. Edge Feature for SG-FIM construction:** Table 10 shows that using both node and edge features for SG-FIM construction significantly outperforms using either feature alone. Node features capture local joint characteristics, while edge features encode relational information between joints. Combining both provides a more comprehensive understanding of the skeleton

Table 8: Effect of in-distribution knowledge retention regularization.

| In-distribution knowledge retention regularization | 200ms | 400ms | 800ms | 1000ms |
|:---:|:---:|:---:|:---:|:---:|
| no | 32.5 | 44.2 | 83.7 | 140.2 |
| yes | 24.6 | 38.7 | 76.8 | 123.1 |

Table 9: Effect of ID-informative subgraph v.s. ID label for Fisher Information Matrix construction.

| ID-Informative Subgraph | ID label | 200ms | 400ms | 800ms | 1000ms |
|:---:|:---:|:---:|:---:|:---:|:---:|
| × | ✓ | 26.8 | 40.4 | 80.1 | 129.5 |
| ✓ | × | 24.6 | 38.7 | 76.8 | 123.1 |

Table 10: Effect of using node feature, edge features, or both, of the subgraph, for SG-FIM construction.

| Node | Edge | 200ms | 400ms | 800ms | 1000ms |
|:---:|:---:|:---:|:---:|:---:|:---:|
| × | ✓ | 25.7 | 59.3 | 78.4 | 126.0 |
| ✓ | × | 27.0 | 60.1 | 79.4 | 129.5 |
| ✓ | ✓ | 24.6 | 38.7 | 76.8 | 123.1 |

structure, resulting in better identification of ID-specific parameters, thus facilitating the ID-specific knowledge preservation. This confirms that the joint use of these nodes and edges in the compressed subgraph is essential for achieving accurate and robust predictions.

**Self-supervised Losses:** Table 11 indicates that using both spatial and smoothness losses together significantly improves prediction performance compared to using each loss independently. The spatial loss ensures that bone lengths between observed and predicted poses remain consistent, preserving the structural integrity of the human skeleton and preventing unrealistic predictions. On the other hand, the smoothness loss enforces temporal consistency by minimizing sudden changes between consecutive frames, which is crucial for generating natural and coherent motion sequences. When combined, these two losses complement each other by capturing both the spatial relationships between joints and the temporal evolution of poses, resulting in more accurate and stable predictions. This comprehensive approach effectively reduces MPJPE, confirming that the joint use of spatial and smoothness losses is essential for achieving robust performance, particularly in scenarios with mixed ID and OOD data.

**Hyperparameters Analysis of $\alpha$ and $\beta$:** This ablation study aims to evaluate the impact of different hyperparameter settings on the performance of our IDKR method. Specifically, we analyze the effect of varying the $\alpha$ parameter, which controls the importance of the Graph Information Bottleneck (GIL) loss, and the $\beta$ parameter, which dictates the weight of the IDKR regularization term. The objective is to find the optimal combination of $\alpha$ and $\beta$ that provides the best balance between capturing informative subgraph features and preserving ID-specific knowledge during adaptation. The results in Table 12 shows that using $\alpha = 0.3$ and $\beta = 0.2$ achieves the best performance. A higher $\alpha$ excessively filters information, potentially losing critical ID features, while a lower $\alpha$ is insufficient for distinguishing ID from OOD data. Similarly, a high $\beta$ over-constrains the model, reducing its flexibility, whereas a low $\beta$ results in the loss of ID-specific knowledge during adaptation. The selected values of $\alpha = 0.3$ and $\beta = 0.2$ provide the optimal trade-off, leading to improved MPJPE and robust model performance. Proper tuning of $\alpha$ and $\beta$ is essential for maximizing the effectiveness of IDKR, enabling it to retain crucial ID knowledge while adapting to OOD data. This configuration ensures optimal model performance across various scenarios.

**Effect of the Time-weight Parameter $\gamma$:** In the context of human motion prediction, the time-weight parameter $\gamma$ plays a crucial role in determining the relative importance of different frames in the observed sequence. Since human motion prediction is inherently a time-series task, it is important to assign greater weight to more recent frames, which carry more relevant information for predicting near-future poses. Conversely, earlier frames, though still useful for providing context, should contribute less to the prediction.

Table 11: Effect of the supervised losses.

| $\mathcal{L}_{\text{TWS}}$ | $\mathcal{L}_{\text{TWT}}$ | 200ms | 400ms | 800ms | 1000ms |
|---|---|---|---|---|---|
| $\times$ | $\checkmark$ | 27.0 | 59.5 | 80.0 | 127.8 |
| $\checkmark$ | $\times$ | 25.2 | 58.3 | 78.1 | 125.6 |
| $\checkmark$ | $\checkmark$ | 24.6 | 38.7 | 76.8 | 123.1 |

Table 12: Investigation of the impact of different hyperparameter settings on the performance.

| $\alpha$ | $\beta$ | 200ms | 400ms | 800ms | 1000ms |
|---|---|---|---|---|---|
| 0.25 | 0.2 | 26.4 | 41.1 | 79.4 | 126.0 |
| | 0.15 | 25.3 | 38.4 | 78.5 | 124.7 |
| **0.3** | **0.2** | 24.6 | 38.7 | 76.8 | 123.1 |
| | 0.25 | 25.0 | 40.1 | 77.7 | 125.2 |
| 0.35 | 0.2 | 24.5 | 39.2 | 77.3 | 124.8 |

Table 13: Effect time-weight parameter $\gamma$.

| $\gamma$ | 200ms | 400ms | 800ms | 1000ms |
|---|---|---|---|---|
| 0.6 | 26.1 | 42.1 | 82.5 | 127.3 |
| 0.7 | 24.6 | 38.7 | 76.8 | 123.1 |
| 0.8 | 24.9 | 40.1 | 79.2 | 126.5 |

By applying an exponential weighting function controlled by $\gamma$, we modulate the impact of each frame in the sequence on the model's loss calculation. The exponential nature of the weighting ensures that smaller values of $\gamma$ distribute the importance more evenly across frames, while larger values of $\gamma$ sharply increase the focus on recent frames. Therefore, the choice of $\gamma$ has a direct impact on how the model balances short-term accuracy with long-term stability. As shown in Table 13, at $\gamma = 0.7$, the model achieves the best performance. This value of $\gamma$ provides the ideal trade-off between prioritizing recent frames and retaining sufficient context from earlier frames. The MPJPE for short-term predictions reached its lowest value, indicating that the model was highly accurate in capturing recent motion dynamics, while maintaining stability in longer sequences. This suggests that at $\gamma = 0.7$, the model effectively balances immediate responsiveness to recent frames with the ability to leverage the broader temporal context.

## G  LIMITATION

The proposed IDKR model is designed to tackle the distribution shift challenges in human pose prediction tasks, particularly in scenarios involving a mixture of ID and OOD data. Our approach is grounded in the concept of retaining ID-specific knowledge during continual Test-Time Adaptation (TTA). This is achieved by compressing the most informative subgraph relative to the in-distribution for any given target skeleton sequence. The methodology behind our IDKR model involves extracting the invariant subgraph from the original human skeleton graph and utilizing this ID-informative subgraph to identify the most relevant parameters for in-distribution sequences.

To effectively preserve ID-specific knowledge, the model incorporates a gradient computation process, which introduces some computational overhead. Specifically, on a single NVIDIA RTX 4090 GPU with PyTorch 2.1.0, our model requires approximately 121ms to adapt to a new target sequence. This computational overhead is relatively small, allowing our method to achieve state-of-the-art performance in terms of Mean Per Joint Position Error (MPJPE). Although this adaptation time is slightly longer than the standard HPP method—siMLPe (84.3ms), it is faster than the state-of-the-art H/P-TTP method (130ms). Therefore, we believe that the proposed method is still efficient enough for practical applications, taking only 121 ms to predict a 1000 ms motion sequence.

