# OpenReview forum: "Never Forget the Basics: In-distribution Knowledge Retention for Continual Test-time Adaptation in Human Motion Prediction"
_ICLR.cc/2025/Conference — ICLR 2025 Conference Withdrawn Submission_

### Official Review · Reviewer_kLDa · 2024-10-27

**Soundness:** 2
**Presentation:** 2
**Contribution:** 2
**Rating:** 3
**Confidence:** 4

**Summary:**

The paper introduces a framework called In-Distribution Knowledge Retention (IDKR) to tackle test-time adaptation (TTA) for human pose prediction (HPP) in dynamic environments. Typically, TTA methods adapt models to out-of-distribution (OOD) data but fail to maintain performance on in-distribution (ID) data, leading to issues when ID and OOD data coexist during inference. In order to address this issue, the key idea of the IDKR framework is to preserve ID-specific knowledge while adapting to OOD data.

The authors propose representing human pose sequences as a graph, where the IDKR framework identifies and retains ID-relevant information through an ID-informative subgraph learning technique. IDKR then computes a Structural Graph Fisher Information Matrix (SG-FIM) based on both node and edge features within this subgraph, to identify the model parameters which are essential for ID data. Two self-supervised losses are finally introduced, allowing the model to adapt effectively to new data while minimizing deviation in ID-specific parameters.

Experiments are conducted on 3 datasets: Humans3.6M, CMU MoCap, and GRAB. Results are shown across multiple scenarios (different subjects/motion/datasets) including cases where ID data is present during testing.

**Strengths:**

1. By specifically addressing the coexistence of ID and OOD data during adaptation, the paper tackles a practical gap in current TTA methods for HPP, which often focus solely on OOD adaptation.

2. Experiments demonstrate the method's effectiveness across multiple settings, showing clear performance improvements over state-of-the-art methods.

**Weaknesses:**

**Method**

1. The exact details of ID-subgraph extraction is unclear. The authors mention that it is implemented by learning binary masks which are then used to mask the over the node and adjacency matrices. There is no mention of how binary constraints are imposed on these masks.
2. In equation 3 the masks are learned by minimizing a binary classification loss where all instances have the same label - why will the masks be incentivized to learn anything?
3. In relation to equation 3, the posterior of the subgraph $q(\mathcal{Z}|\mathcal{G})$ is not defined.
4. Section 3.3 discusses Fisher information, but equation 4/5 calculate it w.r.t the inputs (node and edge features) and not the parameters. How do the dimensions match up? How exactly are these node and edge features calculated?
5. The authors introduce two self-supervised losses which are weighted according to an exponential function, with higher weights for larger timestamps (or more recent frames). Equation 9 defines the time-weighted spatial loss which calculates the consistency of predicted bone lengths across frames - why does the timestamp play a role here given that bone length should remain consistent across time? Similar argument holds for equation 10. Overall the impact is unclear.


**Clarity**

I found the paper to be lacking in clarity.


**Experiments**

Going from the setup $S^+$ to $S^{+-}$ and from $C^+$ to $C^{+-}$ the difference in performance is consistent for both the baseline H/P-TTP and IDKR. This implies that both methods are handling ID data similarly - is the claim that IDKR is better at mixed TTA valid?

**Questions:**

1. Please make the learning process clearer.
- What are the masks?
- How do you define the distribution over graphs?
- How is the Fisher Information computed, especially, what are the node and edge features?

2. Ablation study having no weights in the losses.

---

> ### Author Response · Authors · 2024-11-28
> **Response (Part-1) to Reviewer #kLDa**
>
> ## A1: How to get ID-subgraph
> Thank you for your insightful question regarding the details of ID-subgraph extraction. The "mask" mentioned in the paper is a formalized concept to describe the subgraph extraction process. In our implementation, the mask ${M}$ is not explicitly learned or generated. Instead, the subgraph $Z$ is directly optimized during training. This approach aligns with the standard practices in Graph Information Bottleneck (GIB) methods, such as GOODAT, where subgraphs are learned by optimizing their distributions rather than explicitly computing masks. Below, we provide a detailed explanation:
>
>
> **Formal Definition and Optimization of Subgraphs:**
>
> The subgraph $Z = ({X}_Z, {A}_Z)$ is defined as a subset of the original graph $G = ({X}, {A})$, where ${X}_Z$ and ${A}_Z$ represent the node features and adjacency matrix of the subgraph. While masks ${M}_n$ and ${M}_e$ are formalized representations, they are not explicitly computed or stored.
> Instead, a parameterized neural generator learns the subgraph distribution $p(Z|G)$, directly optimizing the structure and content of the subgraph.
>
> In Graph Information Bottleneck methods (GOODAT [Wang et al., 2024]), it is standard practice to avoid explicitly learning masks. Instead, subgraphs are extracted by directly optimizing $p(Z | G)$, ensuring efficient and flexible implementations. Our approach adheres to this standard.
> > The mask ${M}$ is a formalized representation to describe the subgraph extraction process. In practice, we directly optimize $Z$ through Eq.3 without explicitly learning a mask.  For more explanations, please refer to **A2**.
>
>
> ## A2: What the Eq.3 learning
>
> We clarify that Eq.3 does not involve directly learning explicit masks $M$. Instead, it relies on the Graph Information Bottleneck (GIB) theory, where the subgraph $Z$ is optimized to extract key information. Specifically:
>
> **1st Term in Eq.3:**
>
> The 1st term ensures that the compressed subgraph $Z$ retains the same label as the original graph $G$. This aligns with the GIB principle: a large portion of a graph’s information is irrelevant, and only a small subset of information is critical for the task (e.g., label prediction).
> This term incentivizes $Z$ to focus on label-relevant information, ensuring the subgraph has sufficient predictive capacity.
>
> **2nd Term in Eq.3:**
>
> The 2nd term  uses DL divergence to ensure that there is a high possibility of significant differences between the subgraph and the original data.
> **And, since the original data is a graph structure, the ultimate goal is to force $Z $to be the smallest subset (sub-graph) of the original graph $G$.**
> It prompts the model to discard redundant information and only retain structures highly correlated with the ID data.
>
>
>
> The mask ${M}$ is a formalized representation to describe the subgraph extraction process.
> In practice, we directly optimize $Z$ through Eq.3 without explicitly learning a mask.
> Eq.3 ensure that the subgraph extraction process retains ID label-relevant information while minimizing redundancy. This mechanism is a direct application of GIB theory rather than  mask learning.
>
>
>
>
> ## A3: Posterior distribution in Eq.3
> Thank you for highlighting the concern regarding the posterior definition of the subgraph in Equation 3. Below, we provide a detailed clarification:
>
> **Definition of the Subgraph Posterior:**
> In Eq.3, the posterior distribution of the subgraph $p(Z|G)$ is parameterized through a neural generator. The generator takes the original graph $G = (X, A)$ as input and outputs the conditional distribution of the subgraph $Z$.
> Formally, $p(Z|G)$ is defined as:
>
> $
> p(Z \mid G)=\prod_{i \in V} p (z_i \mid x_i, a_{i,:}),
> $
>
> where $z_i$ represents whether a node or edge is included, $x_i$ is the node feature, and $a_{i,:}$ represents the adjacency information for the node.
>
>
> **Implementation of the Generator:**
>
> To model $p(Z|G)$, we use a neural generator based on a multilayer perceptron (MLP) as in GraphCL (You et al., 2020):
> Node Distribution: $p(z_i|x_i,a_{i,:})$ models the probability of including a node based on its features and adjacency.
> Edge Distribution: $p(z_{ij}|x_i, x_j, {a}_{i,j})$ models the probability of including an edge based on the features of the connected nodes.
> This approach allows flexible modeling of $p(Z|G)$ and ensures that the posterior can adapt well to different scenarios.
>
>
>
> The posterior $p(Z|G)$ is trained to maximize the relevance of the subgraph to the label $Y$ (the first term) while aligning with the prior distribution $\phi(Z)$ (the second term).
> The prior $\phi(Z)$ is typically a sparse Gaussian to ensure the extracted subgraph is compact and task-relevant.
>
> We acknowledge that the explicit definition of $p(Z|G)$ was not clearly stated in the manuscript, and we will address this omission in the revised version.

---

> ### Author Response · Authors · 2024-11-28
> **Response (Part-2) to Reviewer #kLDa**
>
> ## A4: Fisher information matrix in Eq.4/5
> Thank you for your thoughtful question regarding the dimensional alignment in Fisher Information computation in Equations 4 and 5. Below, we provide a detailed explanation, addressing both how the dimensions match up and how the node and edge features are calculated.
>
> **Node and Edge Feature Calculations**
>
> Node Features $X$ comprises:
> (1) Spatial position: 3D coordinates (joint positions).
> (2)Kinematic properties: Velocity, acceleration, or other motion-related attributes.
>
>
> Edge Features $E$ represent the relationships between connected nodes (e.g., parent-child joints). Each edge feature vector includes: (1) Bone length: Euclidean distance between connected nodes.
> (2) Angular information: Angles formed between edges and other connected bones.
>
>
> **Fisher Information w.r.t. Inputs (Eq.4/5)**
>
> In Sec-3.3, we compute the Fisher Information Matrix (FIM) w.r.t. the input features $X$ (i.e., node and edge features) rather than directly with respect to the model parameters. This is done in order to focus on the information in the input features that is most relevant to the task, such as predicting human poses or other motion-related tasks. The FIM in this context is computed as:
>
> $$
> F_v =
> I({X})=\mathbb{E}_{p(Y \mid X)}[\frac{\partial \log p(Y \mid X)}{\partial X} \frac{\partial \log p(Y \mid X)^T}{\partial X}]
> $$
>
> Here, the gradient $\frac{\partial \log p(Y|X)}{\partial {X}}$ is computed with respect to ${X}\in \mathbb{R}^{P}$ for node features and $\mathbf{E}$ has dimensions $\mathbb{R}^{P}$ for edge features.
> $P$ is the number of model parameters.
>
>
> The resulting FIM, $I(X)$, is a $P \times P$ matrix. This is because we are ultimately interested in understanding the sensitivity of the model with respect to changes in its parameters, which is why the Fisher Information Matrix is computed with respect to the model’s parameters (as they influence the predictions).
>
>
>
> **How the Dimensions Match Up**
>
> The key point of your question is how the dimensions of the FIM align with the model parameters and how we compute the gradients. In our approach, we are essentially computing the Fisher Information with respect to the input features, but we still need to link the input features with the model parameters.
>
> Model Parameters $\Theta$:
> The model parameters $\Theta$ have dimensions $P$, i.e., $\Theta \in \mathbb{R}^P$. The Fisher Information Matrix with respect to $\Theta$, denoted $F_\Theta$, typically has dimensions $P \times P$.
>
> Gradient Calculation:
> We compute the gradient $\frac{\partial \log p(Y|X)}{\partial {X}}$ with respect to the input features, which influences the final model parameters. This is done via backpropagation through the network.
>
> Dimension Alignment:
> The gradient $\frac{\partial \log p(Y|X)}{\partial X}$ is computed for each feature in the input (nodes and edges).
> The dimensions of this gradient match with the model parameters $\Theta$, allowing us to compute the FIM and its diagonal version $F_\Theta$ (as in Eq.6), with dimensions $P$ for the full parameter.
> In this case, the $F_\Theta$ is still aligned with the dimensions of the model parameters since the input features are connected through the parameters via the model.
>
> Why Not Directly Compute FIM on $\Theta$?
> Computing FIM directly w.r.t. model parameters can be computationally expensive, especially for high-dimensional models.
> Our method computes it w.r.t. input features, which provides a more efficient way of identifying the task-relevant parts.
> By computing the FIM on input features (nodes and edges), we focus on the parts of the input space that are most relevant to the model’s task.
>
>
>
>
> ## A5: Time-weight in both self-supervised losses
> We use time-weighting in the losses to address the increasing uncertainty in long-term predictions. Here is a more concise explanation:
>
> - **Why Time-Weighting is Used:**
> While bone lengths remain consistent, predicting human poses over long time horizons becomes increasingly uncertain due to the natural variability in human motion. Short-term predictions (within 1s) are more reliable, whereas longer-term tend to accumulate errors.
>
> - **Impact of Time-Weighting:**
> By applying higher weights to recent frames, we focus the model on short-term, more reliable predictions, where pose dynamics are more predictable. Longer timestamps are given lower weights, allowing the model to avoid overfitting to long-term predictions that are less reliable due to the increasing uncertainty.
>
> - **Effect of two Losses:**
> In Eq.9 (spatial loss), this time-weighting ensures that bone lengths, although invariant, are aligned more accurately in short-term frames. In Eq.10 (temporal loss), it strengthens the consistency of nearby frames while reducing the penalty for distant frames, where the prediction uncertainty is higher.
>
> This mechanism helps improve performance by prioritizing more reliable, short-term pose predictions and reducing the impact of long-term prediction errors.

---

> ### Author Response · Authors · 2024-11-28
> **Response (Part-3) to Reviewer #kLDa**
>
> ## A6: IDKR's Advantage in Mixed TTA
>
> We understand your concern that both IDKR and H/P-TTP show similar performance trends in the S+- and C+- settings. However, we would like to clarify why we believe IDKR still provides benefits in mixed TTA scenarios:
>
>
> - Although both methods show consistent performance trends across the S+- and C+- setups, the key distinction lies in **IDKR's ability to maintain consistent performance on ID data** while adapting to OOD sequences. In the mixed TTA scenarios (S+- and C+), **IDKR minimizes performance degradation on ID data**, a critical feature when both ID and OOD data coexist in dynamic environments. This is demonstrated in **Table 5 and Table 6** of Appendix, where IDKR consistently outperforms the baseline H/P-TTP method for ID samples, showing superior accuracy and robustness.
>
>
> - Performance Consistency on ID Data:
> In the separated results of  **Table 5 and Table 6**, IDKR consistently surpasses H/P-TTP in terms of performance on ID samples, demonstrating its ability to better preserve ID knowledge without sacrificing adaptation to OOD data. This performance consistency is a significant advantage, especially in scenarios where the ID data is critical for reliable human pose prediction.
>
> In conclusion, while the performance trends may be similar across both methods, **IDKR's ability to simultaneously handle ID and OOD data** while preserving the ID data's accuracy sets it apart and provides a substantial improvement in robustness, particularly for long-term pose prediction tasks.
>
> We hope this explanation clarifies the unique strengths of IDKR in mixed TTA settings. Please feel free to share any additional questions or suggestions.
>
> ## A7: Response to other questions
>
> **Learning Process:**
>
> We understand that the learning process might not have been clear enough, and we apologize for any confusion caused. To clarify:
>
> Subgraph Learning: We have already explained that the "mask" $\mathbf{M}$ is a formalized concept for describing subgraph extraction, but it is not explicitly learned or generated. Instead, the subgraph $Z$ is directly optimized during training. This approach is aligned with standard practices in Graph Information Bottleneck (GIB) methods, like GOODAT. For further clarification, please refer to A1 ("How to get ID-subgraph") and A2 ("What the Eq.3 learning") in our response.
> Code Availability: In addition to further clarifying the method in the revised paper, we will make our code publicly available to provide complete transparency and allow others to replicate and better understand the learning process. This will help readers explore the implementation details and ensure reproducibility.
> We believe that by providing this additional resource, we will improve the accessibility and clarity of our approach.
>
>
>
> **Other Related Questions:**
>
> We understand that there were some additional questions about the definition of masks, the posterior distribution in Equation 3, the computation of the Fisher Information Matrix, and the impact of time-weighting in the losses. We have already addressed these questions in detail in previous sections:
>
> - **Masks:** The "mask" is a formal concept to describe subgraph extraction. We do not explicitly learn or compute the mask but optimize the subgraph directly through Equation 3. For more details, please see **A1** and **A2**.
>
> - **Posterior Distribution in Eq3**: The posterior $p(Z|G)$ is parameterized via a neural generator. For a full explanation of how this is computed, please refer to **A3** ("Posterior distribution in Eq.3").
>
> - **FIM**: The Fisher Information is computed with respect to the input features $\mathbf{X}$ (node and edge features), and we clarified how the dimensions match up and how node and edge features are calculated in ** A4**.
>
> - **Time-Weighting in Losses:** We addressed the role of time-weighting in the spatial and temporal losses in **A5**("Time-weight in both self-supervised losses").
> -
> We hope that these clarifications, along with the additional code release, will address the concerns raised regarding clarity and the learning process.
>
> **Note:**
> We sincerely appreciate the reviewers' feedback, which has helped us improve the quality and clarity of our work. In addition to the revisions, we will ensure a thorough proofreading of the manuscript to improve its expression. The code will also be made publicly available to enhance transparency and reproducibility.
>
> Thank you again for your valuable suggestions.

---

> ### Author Response · Authors · 2024-12-02
> **Request for Further Discussion on Review Comments**
>
> I am one of the authors of this manuscript. First and foremost, I would like to express our sincere gratitude for your attention to our work and the valuable comments provided during the initial review process.
>
> We have prepared a detailed rebuttal in response to the preliminary review comments and are eager to engage in further discussion to address any concerns or questions you may have. Since submitting our rebuttal, we have been looking forward to receiving your feedback but have not yet received an update.
>
> We highly value your insights and believe that through additional dialogue, we can refine and improve our manuscript. If it is convenient for you, we kindly request your feedback in any form—whether it be questions regarding our rebuttal, suggestions, or requests for more information.
>
> Please let us know how we can best assist in facilitating this process. Should you prefer to discuss specific iss

---

### Official Review · Reviewer_1WGo · 2024-11-03

**Soundness:** 3
**Presentation:** 3
**Contribution:** 2
**Rating:** 5
**Confidence:** 4

**Summary:**

This paper frames the Human Pose Prediction (HPP) task in a continual-Test Time Adaptation (C-TTA) scenario where it accounts for maintaining the performance on both ID and OOD data, which it argues matches the target distribution and  to prevent model degradation on ID sequences while selectively adapting to OOD data. It employs a subgraph learning strategy and a Structural Graph Fisher Information Matrix (SG-FIM) to preserve essential parameters for ID data by leveraging the structural consistency in human skeletons. Experiments are shown on the Humans 3.6m, CMU-Mocap and GRAB datasets and showed improvement when compared to 3 other TTA and C-TTA methods.

**Strengths:**

1. Using Fischer Information as a regularizing technique is an interesting way of approaching the problem of HPP under Continual Test Time Adaptation.
2. The experiments conducted are comprehensive and covers a wide set of works not just related to HPP but also from traditional TTA literature like TENT, CoTTA etc.

**Weaknesses:**

1. The paper looks incremental in nature: there is limited novelty in the proposed HPP setting. The motivation of the paper seems to come from that fact that there is continuous change in the target domain. When a person performs some activity, it is fair to assume that the environment does not change drastically which may bring in significant domain shift (rain, fog etc.). Some plots in Figure 3 (S+- and C+-) seems to illustrate this.

2. The fischer information matrix designed here is essentially able to act as a regularizer since it serves as a prior of human anatomy (ex: invariant bone lengths). Several previous works exist in Human Pose Estimation literature which propose strong priors to capture a plausible set of human poses. It seems FIM was chosen as a straight-forward choice - there is a possibility this method may fail in scenarios where there are significant pose variations/self-occlusons due to heavy reliance on just invariant features. In scenarios where poses vary significantly even within the ID distribution (e.g., differing postures or activities within the same subject), SG-FIM’s reliance on just invariant features may lead to suboptimal adaptation as it overemphasizes preserving parameters linked to structural constraints rather than flexible, adaptive knowledge which is the main motivation behind any C-TTA setting.

3. The loss terms in 9 and 10 don't particularly seem novel: test-time alignment of human poses based on spatial and temporal features is a well-known technique. I would like to see if this was introduced based on some specific enhancement in mind to complement the GIL Learning approach or just as additional regularizers between frames.

4. The authors introduce specific setups like Setup-C+− and Setup-S+− to simulate mixed ID and OOD domains by including only 10% of the source domain data as ID in the target domain. However, these ratios look somewhat arbitrary - testing with different ID-to-OOD ratios would provide more insight.

4. Typos present: 800.9 PCK@150 in Table 3.

If the authors can address some of these concerns, then I believe it would significantly increase the paper's relevance.

**Questions:**

Please refer to the weaknesses section.

---

> ### Author Response · Authors · 2024-11-27
> **Response (Part-1) to Reviewer #1WGo**
>
> ### A1: Domain shift instable environments
> While environmental factors (e.g., rain, fog) are often associated with domain shifts, in human motion capture tasks, domain shifts can arise even when the external environment remains stable:
>
> - **Variations in Individual Motion Pattern:**  Even in the same environment, biomechanical differences among individuals (e.g., joint flexibility, stride length, or motion speed) can result in noticeable prediction challenges. For instance, training data dominated by adults may lead to degraded performance when tested on teenagers or elderly individuals.
> - **Gradual Changes in Behavior:** Over time, a person performing the same task may show changes in their movements due to fatigue, increased proficiency, or random variations, leading to a shift in the data distribution.
> - **Temporal Dependency Issues:** Prediction of future poses relies on past motion patterns. Sudden anomalies or deviations (e.g., slips, falls) can disrupt this dependency, causing domain shifts.
>
>
>
> Figure 3's t-SNE visualization (S+- and C+-) is intended to illustrate the distributional differences when target domain includes both new subjects (categories) and same subjects (categories).
> The plot highlights that, although subjects or actions, as well as scenes, are similar between source and target domain, the latent space distribution of ID and OOD data can be different,  leading to ID and OOD.
> This supports the premise that domain shifts can occur without obvious environmental changes.
>
> ### A2: On SG-FIM’s Dependence on Skeletal Priors
>
> We understand the concerns regarding the potential limitations of the SG-FIM in scenarios involving significant pose variations and self-occlusions. Below, we address this issue in detail:
>
> While SG-FIM leverages anatomical priors such as invariant bone lengths, it does not entirely rely on these fixed features. Instead, our method incorporates a dynamic subgraph learning process that identifies the most informative structures for ID data. This process adapts to distributional variations within the ID domain, ensuring that even with significant pose diversity, the framework retains the most relevant information.
>
> #### Mechanisms to Handle Significant Pose Variations and Self-Occlusions
> - **Pose Variations:** In cases of substantial pose shifts (e.g., transitions between sitting and standing postures), SG-FIM captures both node (joint) and edge (skeletal connection) features, balancing local flexibility with global consistency. For example, while joint angles may change dramatically, edge features (bone lengths) provide a stable reference, and dynamic node features complement this stability by adapting to pose-specific variations.
> - **Self-Occlusions:**  In self-occlusion scenarios (e.g., crossed arms or complex movements), the binary masks generated during subgraph learning automatically exclude irrelevant or noisy nodes and edges. This reduces the impact of uncertain data, enhancing the model’s robustness.
>
> #### Comparison to Other Strong Priors
> - Our approach differs fundamentally from traditional pose estimation methods that rely on fixed skeletal priors (e.g., template-based skeletal models):
>
> - SG-FIM integrates dynamic subgraph learning, which does not constrain the model to static structural assumptions but instead learns and preserves key parameters contributing to the ID distribution.
> Moreover, edge features in SG-FIM are not static; they are progressively refined during TTA, allowing the model to remain adaptable to OOD scenarios while maintaining ID performance.

---

> ### Author Response · Authors · 2024-11-27
> **Response (Part-2) to Reviewer #1WGo**
>
> ### A3: Eq.9 and Eq.10
>
> - We agree that that these losses—spatial  loss and temporal alignment loss—time-weighted spatial loss and temporal loss—are not novel designs, and we do not claim them as a major contribution of our work. Instead, our primary enhancement lies in the introduction of **time-weighting mechanisms**:
>
>   > **Role of Time-Weighting Mechanisms:** In HPP, model accuracy often deteriorates significantly as the prediction horizon increases, which is particularly evident in long-term sequence forecasting. To address this issue, we incorporate time-weighting into the spatial loss (Eq. 9) and temporal loss (Eq. 10), assigning greater weights to farther time steps. This forces the model to focus more on aligning distant frames during training, thereby improving its performance in long-term predictions.
>
>  - **How Time-Weighting Complements:**
>   *Spatial Alignment Loss:* Skeletal structures (e.g., bone lengths) are inherently invariant over time. However, prediction errors tend to accumulate in longer time horizons. By incorporating time-weighting, we reinforce the alignment of distant frames, ensuring the stability of skeletal structures in long-term predictions.
> *Temporal Alignment Loss:* Predicting long-term sequences requires capturing dynamic changes across consecutive time steps, but distant frames are more prone to deviation from true trajectories. The time-weighted design strongly constrains the dynamic consistency of distant frames, improving the model’s robustness in long-term sequence forecasting.
>
>
> In Appendix-F, we present ablation studies where the time-weighting mechanism is removed. Results show a noticeable degradation in MPJPE, particularly in long-sequence prediction scenarios. It evidences the importance for improving long-term prediction performance in dynamic target domains.
>
>
> ### A4: Ratio of ID samples
> Among all our experimental setups, only the artificially constructed setups (Setup-C+− and Setup-S+−) use a 10% ID sample ratio. This choice was made to provide a controlled benchmark for evaluating the model’s performance in mixed-distribution scenarios. The 10% ratio was selected to simulate target domains where OOD/ID data predominates, while retaining a certain proportion of ID data to assess the model’s behavior under mixed conditions. As an intermediate value, 10% strikes a balance for testing the model's adaptability across varying proportions.
>
>
> **Additional Experiments with Varying Ratios:**
> To further validate the robustness of the IDKR framework under different ID-to-OOD ratios of GRAB dataset, we conduct additional experiments with ID ratios of 5%, 10%, 15%, and 20%.
>
> | Ration of Source        | 200ms | 400ms | 600ms | 1000ms |
> |-|-|-|-|-|
> | 5%               | 27.9 | 51.5  | 88.4 | 136.7 |
> | 10%  (original)  |27.1  | 49.3  | 85.2 | 130.5|
> | 15%              | 26.8 | 48.2  | 82.2 | 128.9 |
> | 20%              | 26.5 | 46.5  | 83.6 | 127.7 |
> **Table-r2:** MPJPE for Setup-S$^{+-}$ of Mixed-Distributions
>
> | Ration of Source        | 200ms | 400ms | 600ms | 1000ms |
> |-|-|-|-|-|
> | 5%               | 32.9 | 51.5  | 88.2 | 139.3 |
> | 10%  (original)  | 31.5 |50.3 |88.0 |137.5|
> | 15%              | 30.4 | 51.2  | 86.4 | 134.0 |
> | 20%              | 20.7 | 49.7  | 85.7 | 132.3|
> **Table-r3:** MPJPE for Setup-C$^{+-}$  of Mixed-Distributions
>
>
>
> These additional experiments further confirm that IDKR not only enhances ID performance across varying ID-to-OOD ratios but also significantly reduces prediction errors at longer horizons (e.g., 600ms and 1000ms).
> Across the tested range of ID ratios (5%-20%), IDKR consistently delivers performance improvements. This demonstrates the framework’s robustness in handling a wide range of mixed distribution scenarios.
> The stability across different ID-to-OOD ratios reflects the effectiveness of the SG-FIM approach and the adaptability of the time-weighted losses to dynamic scenarios.
> The results underscore the strong adaptability and robustness of our approach to mixed-distribution conditions, regardless of changes in ID proportions.
>
>
>
> ### A5 Typos:
> The correct text of '800.9' should be '80.9'.
>
> We greatly appreciate the reviewers’ constructive feedback. In addition to the revisions mentioned, we will ensure a thorough proofreading of the paper to further improve its clarity and accuracy. Thank you again for your valuable suggestions.

---

> ### Author Response · Authors · 2024-12-02
> **Request for Further Discussion with Review 1WGo**
>
> I am one of the authors of this manuscript. I would like to express our sincere gratitude for your attention to our work and the valuable comments provided during the initial review process.
>
> We have recently submitted a detailed rebuttal addressing the comments, and we are now eager to engage in further discussion with you to address any concerns or questions you may have. Specifically, we have already responded to several key points, including:
>
> 1. **Domain Shifts in Stable Environments**: We discussed how variations in individual motion patterns, gradual changes in behavior, and temporal dependencies can cause domain shifts even when the external environment remains stable.
> 2. **SG-FIM’s Dependence on Skeletal Priors**: We explained how our method leverages dynamic subgraph learning to handle significant pose variations and self-occlusions.
> 3. **Time-Weighting Mechanisms**: We detailed the role of time-weighting in improving long-term prediction performance.
> 4. **Ratio of ID Samples**: We presented additional experiments with varying ID-to-OOD ratios to demonstrate the robustness of our model.
>
> We have prepared a detailed rebuttal document that covers these points and more, and we are looking forward to receiving your feedback. Your insights are invaluable to us, and we believe that through further dialogue, we can refine and improve our manuscript.
>
> Since submitting our rebuttal, we have not yet received an update and are eager to hear your thoughts. If it is convenient for you, we kindly request your feedback in any form—whether it be questions regarding our rebuttal, suggestions, or requests for more information.
>
> Please let us know how we can best assist in facilitating this process. Should you prefer to discuss specific issues directly or require additional materials from us, we are fully committed to cooperating.
>
> Thank you very much for your time and effort dedicated to our work. We look forward to hearing from you soon and are eager to engage in further discussions.

---

### Official Review · Reviewer_MLBA · 2024-11-04

**Soundness:** 3
**Presentation:** 4
**Contribution:** 3
**Rating:** 6
**Confidence:** 3

**Summary:**

The paper introduces a novel approach to Test-Time Adaptation (TTA) for human pose prediction that addresses real-world scenarios where both in-distribution (ID) and out-of-distribution (OOD) motion data appear in the deployment domain. To manage this dual data presence, the authors propose an In-Distribution Knowledge Retention (IDKR) mechanism, designed to retain crucial ID information while adapting to OOD sequences. The method leverages the Graph Information Bottleneck framework to extract the most informative subgraph for ID data, filtering out irrelevant elements and enhancing focus on essential structures. A Structural Graph Fisher Information Matrix (SG-FIM) is then employed to identify key model parameters that contribute to ID sequence prediction. These parameters are selectively preserved during adaptation through a regularization term integrated into the TTA optimization, ensuring robust ID performance amidst OOD adaptation. Experimental results indicate that IDKR outperforms existing methods in diverse real-world scenarios, underscoring its effectiveness and applicability.

**Strengths:**

1. Novel Framework for Mixed ID and OOD Data in Human Pose Prediction (HPP): The paper introduces a pioneering framework that explicitly addresses the coexistence of in-distribution (ID) and out-of-distribution (OOD) data in human pose prediction tasks, a novel scenario largely overlooked by existing test-time adaptation (TTA) methods.
2. Innovative In-Distribution Knowledge Retention (IDKR) Mechanism: The proposed IDKR mechanism is a standout contribution, using ID-informative subgraph learning combined with a Structural Graph Fisher Information Matrix (SG-FIM). This approach identifies and retains key ID-specific parameters while adapting to OOD data, ensuring that critical ID information is preserved during adaptation. This dual capability of adaptation and retention is a novel approach that improves robustness and accuracy in complex, mixed-domain scenarios.
3. Extensive and Competitive Evaluation: The paper provides thorough experimental evaluations, demonstrating that IDKR significantly outperforms state-of-the-art HPP models, particularly in non-stationary environments with both ID and OOD data. This strong empirical performance indicates the method’s effectiveness and potential for future HPP tasks that encounter dynamic and evolving domain shifts.

**Weaknesses:**

The authors propose a TTA approach that aims to address a “realistic” scenario where both ID and OOD data coexist in dynamic target domains. However, the paper could benefit from more concrete real-world examples and specific use cases to demonstrate the relevance and feasibility of this framework. Adding examples of real-world applications where dynamically shifting ID and OOD data are common, would strengthen the justification for this method and better highlight its practical impact.

The concept of domain shifts is also somewhat underdeveloped in the context of skeletal data, where inputs are limited to pose estimates. The paper would be clearer with a more explicit definition of domain shifts, including measurable criteria for detecting these changes within skeletal data and an explanation of how these shifts affect model performance. Clarifying how the model differentiates ID and OOD data based solely on skeletal data, or whether additional information sources (like RGB image) would be beneficial, would help reinforce the proposed approach’s robustness in real-world scenarios.

Another area for enhancement is in addressing the inherent variability of future pose predictions. Predicting future skeletal positions involves multiple potential outcomes, but the paper approaches predictions deterministically. To provide a fuller picture of the model’s capabilities, discussing how the method might handle multiple plausible future paths could improve its applicability. Additionally, exploring the model’s predictive accuracy over extended time frames and including a graph showing MPJPE across longer intervals would clarify the scope and limitations of the predictions. This would offer more insight into how effectively the model sustains accuracy over time, adding depth to the performance analysis.

The statements on Line 49 and Line 52 reference prior research developments but lack specific citations to support the claims. Adding relevant citations here would provide necessary context.

**Questions:**

1. Real-World Use Cases: Could you provide specific real-world scenarios where the coexistence of ID and OOD data is encountered in dynamic target domains for HPP? Clear examples would help solidify the relevance of the approach.
2. Domain Shift Criteria: How are domain shifts defined and detected solely from skeletal pose data? Could you clarify measurable criteria used for distinguishing domain changes and explain if additional data sources would improve domain differentiation?
3 . Handling Prediction Ambiguity: How does the model manage multiple plausible future poses from skeletal data, given the non-deterministic nature of human motion? Would incorporating probabilistic or multi-path prediction improve the model’s robustness?
4. Long-Term Predictive Accuracy: Could you provide a temporal analysis showing how MPJPE error evolves over extended adaptation periods? A graph of MPJPE over time would clarify the model’s performance in long-term deployments.

# UPDATE (After Discussion Period)

I would like to thank the authors for providing such a detailed response. However there are few points that still remains a matter of concern for me.

- Lack of Explicit Validation for Real-World Relevance (A1): While the authors provide compelling real-world examples, these are largely anecdotal. A stronger case would involve showcasing datasets or use cases where such scenarios are observed, along with empirical validation of the method’s performance in those contexts.

- Scope and Limitations of Deterministic Prediction (A3.1): Although the authors justify focusing on deterministic prediction as the standard for TTA-based HPP, this limits the broader appeal of the paper. Probabilistic approaches, which capture the inherent variability of human motion, are gaining traction in the field. While extending the model to support probabilistic predictions may be beyond the scope of this paper, a discussion of how the proposed framework could adapt to such paradigms in future work would strengthen its contribution.

- Empirical Performance and Long-Term Predictions (A3.2): The inclusion of long-term prediction results is a positive step, but the reported increase in MPJPE over time highlights limitations in handling motion variability for extended horizons. This suggests that while the method outperforms baselines, it may not generalize as effectively for long-term predictions in real-world applications.

While the paper presents a novel and valuable contribution to HPP and the authors have addressed many concerns effectively, **the work does not fully meet the standards of a Strong Accept hence I would like to retain the original rating**.

---

> ### Author Response · Authors · 2024-12-02
> **Request for Further Discussion with Reviewer MLBA**
>
> I hope this message finds you well.
>
> I am the author of this manuscript. I would like to express our sincere gratitude for your attention to our work and the valuable comments provided during the initial review process.
>
> We have carefully reviewed and addressed the concerns and suggestions raised in your initial feedback. To ensure clarity and provide detailed responses, we have prepared a comprehensive rebuttal document covering the following key points:
>
> 1. **Concrete Real-World Examples**: We provided specific examples of ID and OOD data coexistence in real-world applications, such as robots in dynamic environments and surveillance systems.
> 2. **Definition of Domain Shifts in Skeletal Data**: We explained how our model implicitly addresses domain shifts through ID-informative subgraph learning and selective adaptation with the Structural Graph Fisher Information Matrix (SG-FIM).
> 3. **Probabilistic/Deterministic Motion Prediction**: We clarified our focus on deterministic prediction and discussed why this approach is suitable for our goals.
> 4. **Long-Term Prediction Accuracy**: We conducted additional experiments to evaluate our method's performance in long-term prediction tasks and provided detailed results.
> 5. **Missing Citations**: We identified and corrected the missing citations in the manuscript.
>
> We believe these responses provide a clear and thorough explanation of our approach and address the concerns raised. However, we are eager to engage in further discussion to ensure that all questions and concerns are fully addressed. Your insights are invaluable to us, and we are committed to refining our work based on your feedback.
>
> If it is convenient for you, we kindly request your feedback on our rebuttal. Any additional comments or suggestions you may have would be greatly appreciated. We are fully committed to collaborating to improve the quality of our manuscript.
>
> Thank you once again for your time and effort. We look forward to your valuable input.

---

### Note · Authors · 2025-03-05

I have read and agree with the venue's withdrawal policy on behalf of myself and my co-authors.

---

### Meta-Review · Area_Chair_p72d · 2024-12-21

**Metareview:**

The authors have not been able to convince two reviewers (1WGo and kLDa) towards the positive side; these reviewers agreed this work needs extra efforts to reach the acceptance bar of the ICLR. Thus I am inclined towards not accepting this draft at this stage. Thank you for your effort. It is an interesting work. I hope input from the reviewers will help you improve this work further.

**Additional Comments On Reviewer Discussion:**

NA

---

### Decision · Program_Chairs · 2025-01-22

Reject